# Ageing affects DNA methylation drift and transcriptional cell-to-cell variability in mouse muscle stem cells

Irene Hernando-Herraez[1,7], Brendan Evano[2,3,7], Thomas Stubbs [1,6,7], Pierre-Henri Commere[4], Marc Jan Bonder[5], Stephen Clark [1], Simon Andrews[1], Shahragim Tajbakhsh [2,3] & Wolf Reik [1]

Age-related tissue alterations have been associated with a decline in stem cell number and function. Although increased cell-to-cell variability in transcription or epigenetic marks has been proposed to be a major hallmark of ageing, little is known about the molecular diversity of stem cells during ageing. Here we present a single cell multi-omics study of mouse muscle stem cells, combining single-cell transcriptome and DNA methylome profiling. Aged cells show a global increase of uncoordinated transcriptional heterogeneity biased towards genes regulating cell-niche interactions. We find context-dependent alterations of DNA methylation in aged stem cells. Importantly, promoters with increased methylation heterogeneity are associated with increased transcriptional heterogeneity of the genes they drive. These results indicate that epigenetic drift, by accumulation of stochastic DNA methylation changes in promoters, is associated with the degradation of coherent transcriptional networks during stem cell ageing. Furthermore, our observations also shed light on the mechanisms underlying the DNA methylation clock.

[1] Epigenetics Programme, Babraham Institute, Cambridge CB22 3AT, UK. [2] Stem Cells & Development, Department of Developmental & Stem Cell Biology, Institut Pasteur, 25 rue du Dr. Roux, 75015 Paris, France. [3] CNRS UMR 3738, Institut Pasteur, Paris 75015, France. [4] Cytometry and Biomarkers, Center for Technological Resources and Research, Institut Pasteur, 28 rue du Dr. Roux, 75015 Paris, France. [5] European Molecular Biology Laboratory, European Bioinformatics Institute, Wellcome Genome Campus, Hinxton CB10 1SD, UK. [6] Present address: Chronomics Limited. Mills & Reeve Llp, 1 St James Court, Norwich NR3 1RU, UK. [7] These authors contributed equally: Irene Hernando-Herraez, Brendan Evano, Thomas Stubbs. Correspondence and requests for materials should be addressed to I.H.-H. (email: irene.herraez@babraham.ac.uk) or to S.T. (email: shahragim.tajbakhsh@pasteur.fr) or to W.R. (email: wolf.reik@babraham.ac.uk)

Epigenetic alterations have been proposed to be a major cause of age-related decline in tissue function[1]. Changes in DNA methylation are well correlated with ageing, and methylation of specific loci has been used to predict age in a large number of tissues[1,2]. Epigenetic-based age predictive models, also referred to as epigenetic clocks, have been widely applied in the last few years and are believed to reflect chronological aging[3]. Despite their accuracy, the mechanisms underlying these models are largely unknown[2,3].

Notwithstanding the close associations with age, age-related methylation changes are poorly correlated with transcriptional variation, presumably because the changes are generally small and may not occur homogeneously in all cells[2], a phenomenon also known as epigenetic drift. Although epigenetic drift has long been hypothesised to be an important hallmark of ageing[4], this proposal has been challenging to test because of technical constraints. However, powerful combined single cell methods[5,6] are now available, and epigenetic changes during ageing together with their functional consequences can now be read out in single cells[7].

Degenerative changes in tissue-specific stem cells have been proposed to be a major cause of age-related decline in tissue function[8]. While several reports indicate a loss of clonal diversity during early life stages[9–11], little is known about how cell-to-cell variability at the molecular level is involved in stem cell ageing. Here, we performed parallel single-cell DNA methylation and transcriptome sequencing (scM&T-seq) on the same cell[6] to investigate how ageing affects transcriptional and epigenetic heterogeneity of tissue-specific stem cells, using mouse muscle stem cells as a model. Muscle stem cells express the transcription factor Pax7[12] and are largely quiescent in adult muscles. They activate upon injury to differentiate and fuse to form new fibres, or self-renew to reconstitute the stem cell pool[12]. Age-associated muscle defects have been attributed to a decrease in stem cell number together with impaired regenerative potential[13]. In addition, clonal lineage-tracing of mouse muscle stem cells showed that population diversity is unaltered during homoeo-static ageing[14].

Here, by combined single-cell transcriptome and DNA methylome profiling in muscle stem cells, we show a global increase of uncoordinated transcriptional heterogeneity and context-dependent alterations of DNA methylation with age. Notably, old cells that change the most with age reveal alterations in the transcription of genes regulating cell-niche interactions. These findings show linked increases in heterogeneity between the epigenome and the transcriptome, with consequent degradation of coherent transcriptional networks and stem cell functional decline during ageing.

## Results

**Transcriptomic profiling of young and old muscle stem cells**. Muscle stem cells with high expression of Pax7 were shown to be in a deep quiescent, or dormant, state[15,16]. To investigate the molecular effects of ageing in a defined population that is less poised to enter the cell cycle, we isolated single muscle stem cells by fluorescence-activated cell sorting (FACS) from young (1.5 months) and old (26 months) Tg:Pax7-nGFP mice[17] and selected those with high levels of GFP, to which we applied scM&T-seq (Fig. 1a). Importantly, this approach allows us to study variability while minimising key confounder factors such as differences in cell cycle or differentiation states between ages.

After quality control and filtering, a total of 377 transcriptomes from four young and two old mice were analysed. Young (n = 253) and old (n = 124) cells from different individuals clustered together, respectively, indicating no global differences with age and absence of sequencing-related batch effects (Fig. 1b). We also

assigned a cell cycle stage[18] to each cell and observed that, with the exception of one cell from sample Young 2, cells were unlikely to be cycling at the time of isolation, and showed no differences between ages (Supplementary Fig. 1). Measures of cell cycle entry through BrdU uptake in vivo are also in agreement with this observation, supporting the deep quiescent state of these cells (Supplementary Fig. 2). Furthermore, we did not observe significant differences in the levels of Pax7, the myogenic factors Myod and Myf5 and the cell cycle inhibitor Cdkn1b (p27), nor of senescent markers such as Cdkn2a (p16INK4a/p14arf), suggesting that some molecular signatures are conserved between the analysed cell populations (Fig. 1c). Nevertheless, 940 genes were differentially expressed between young and old individuals (SCDE, FDR P < 0.05, Supplementary Data 1, Supplementary Fig. 3), with small differences in some cases; Spry1, which is a key factor for maintaining quiescence[19], and the cell cycle regulators Ccnd1 (Cyclin D1), Btg1 and Gas1 were down-regulated, while ageing markers such as the chemokine genes Ccl11 and Ccl19 were upregulated[16] (Fig. 1c). Furthermore, we identified significant alterations in expression of genes not previously reported to change in expression with age, such as the early activation markers Fosb and Egr1[20] and the metalloproteinase Mmp2 (Fig. 1c).

**Increased transcriptional variability with age**. To investigate if ageing affects transcriptional heterogeneity of the stem cell pool, we calculated pairwise correlation coefficients between cells within each individual (see "Methods") and observed that old individuals showed consistently lower correlation (1.3 mean-fold decrease, Mann–Whitney–Wilcoxon test; P < 2.2e-16, Fig. 1d), indicating a remarkably lower degree of similarity between cells and no obvious population substructure. Similar results were observed when performing the analysis on cohorts of 10 cells from the same individual where the mean correlation in young individuals was always higher than in old ones (P < 0.001) (Supplementary Fig. 4).

We also computed an expression-level normalised measure of gene expression heterogeneity (named distance to the median)[21], which proved to be higher in old individuals (Mann–Whitney–Wilcoxon test; P < 2.2e-16, Fig. 1e), revealing a striking global increase of uncoordinated transcriptional variability with age. Notably, the proportion of cells expressing a given gene (frequency of gene expression) was reduced with age (Mann–Whitney–Wilcoxon test; P < 2.2e-16, Fig. 1f), even for genes that did not significantly change mean expression levels (SCDE, FDR P > 0.05, Fig. 1f). Importantly, we observed that this was independent of gene expression levels and not restricted to lowly expressed genes (Fig. 1g), suggesting that this global feature is not due to technical effects.

Genes that displayed increased expression variability with age (expression frequency difference > 15%) were enriched in extracellular matrix processes and include several collagen genes (Col4a2, Col5a3, Col4a1) and other extracellular matrix-related genes such as Dag1, Sparc, Cdh15, Lamc3 or Itgb1 (Fig. 2a, Supplementary Figs. 5 and 6). Interestingly, muscle stem cells without Itgb1 (β1-integrin) cannot maintain quiescence, and its experimental activation improves ageing-related decline in muscle regeneration[22]. Similarly, reduction of N-cadherin and M-cadherin (Cdh15) leads to a break of quiescence of muscle stem cells[23]. Notably, none of the above-mentioned genes were shown to change in expression level during the isolation procedure of muscle stem cells[24].

To further investigate the increased expression variability with age, we sequenced by scRNA-seq the total population of Pax7 positive cells in four individuals (two young, two old) and

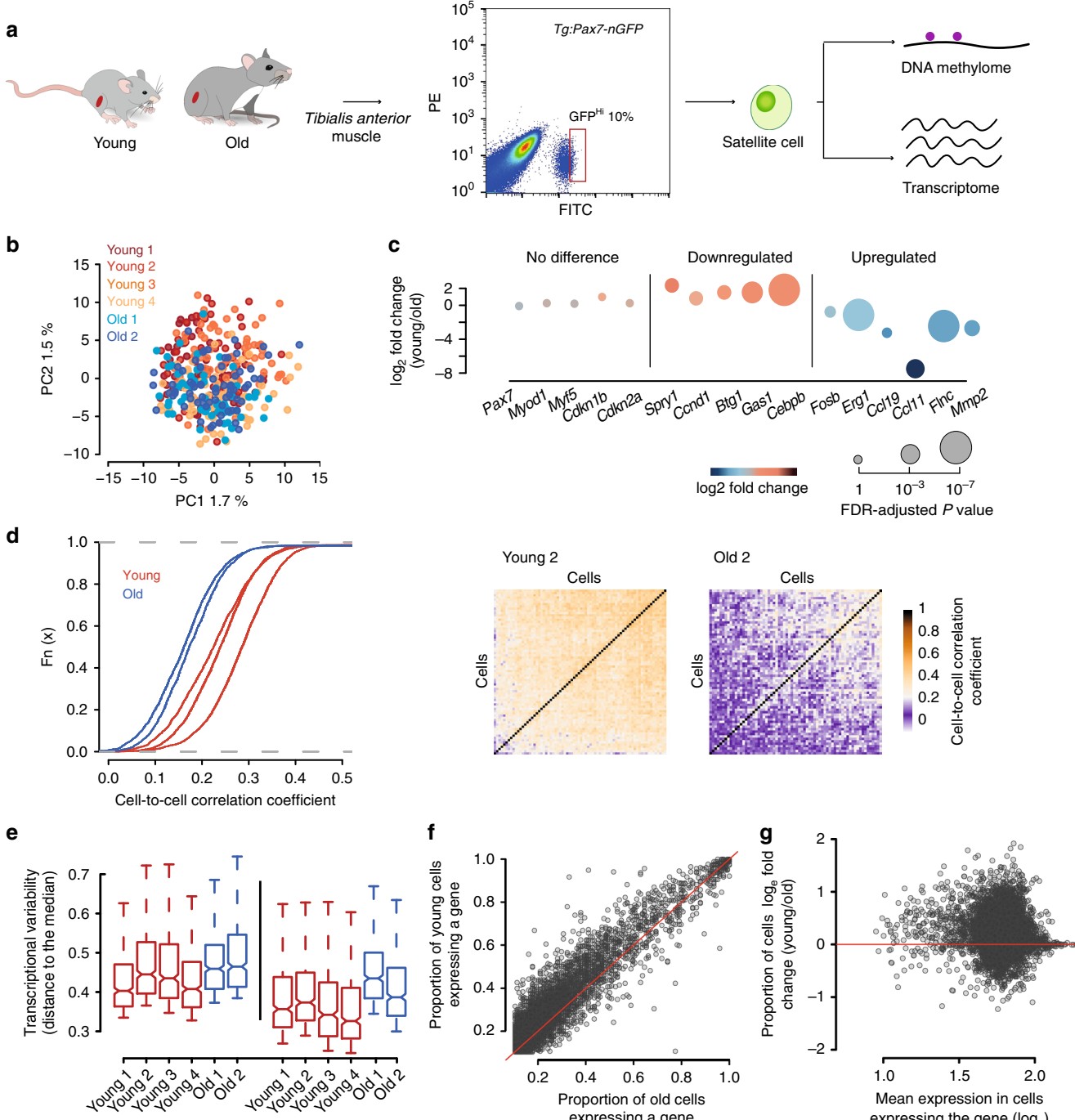

**Fig. 1** Aged muscle stem cells have increased cell-to-cell transcriptional variability. **a** Experimental scheme. Single cells were isolated from *Tg:Pax7-nGFP* young and old mice and subjected to parallel single-cell methylation and RNA sequencing. **b** Principal Component Analysis (PCA) of a total of 377 cells from young ($n = 4$) and old ($n = 2$) individuals. **c** Selected markers and differentially expressed genes between young and old cells. Source data are provided in Supplementary Data 1. **d** Cumulative distribution of cell-to-cell Spearman correlation values per individual (left) showing that transcriptional heterogeneity dramatically increases with age. Heatmap showing cell-to-cell Spearman correlation values from a young and an old mouse (right). **e** Distance to the median of the top 500 most variable genes among all genes (left) and of the top 500 most variable genes among the 5127 common genes expressed in the six individuals (right). For all boxplots, the box represents the interquartile range and the horizontal line in the box represents the median ($n = 500$). **f** Frequency of gene expression in young and old cells. Note that the proportion of cells expressing a given gene is reduced in old cells. **g** Independence between frequency of gene expression differences and gene expression level

observed similar results regarding transcriptional variability (Supplementary Fig. 7). Furthermore, we measured the levels of Cdh15 and Itgb1 proteins in quiescent muscle stem cells from young and old *Tg:Pax7-nGFP* mice after isolation of single myofibres and quantification of immunofluorescence intensity

following immunostaining (Supplementary Fig. 8). Young and old muscle stem cells showed homogeneous nGFP expression, consistent with the absence of difference in *Pax7* expression level between ages as measured by scM&T-seq (Fig. 1c). However, while all the cells analysed expressed Cdh15 and Itgb1 protein in

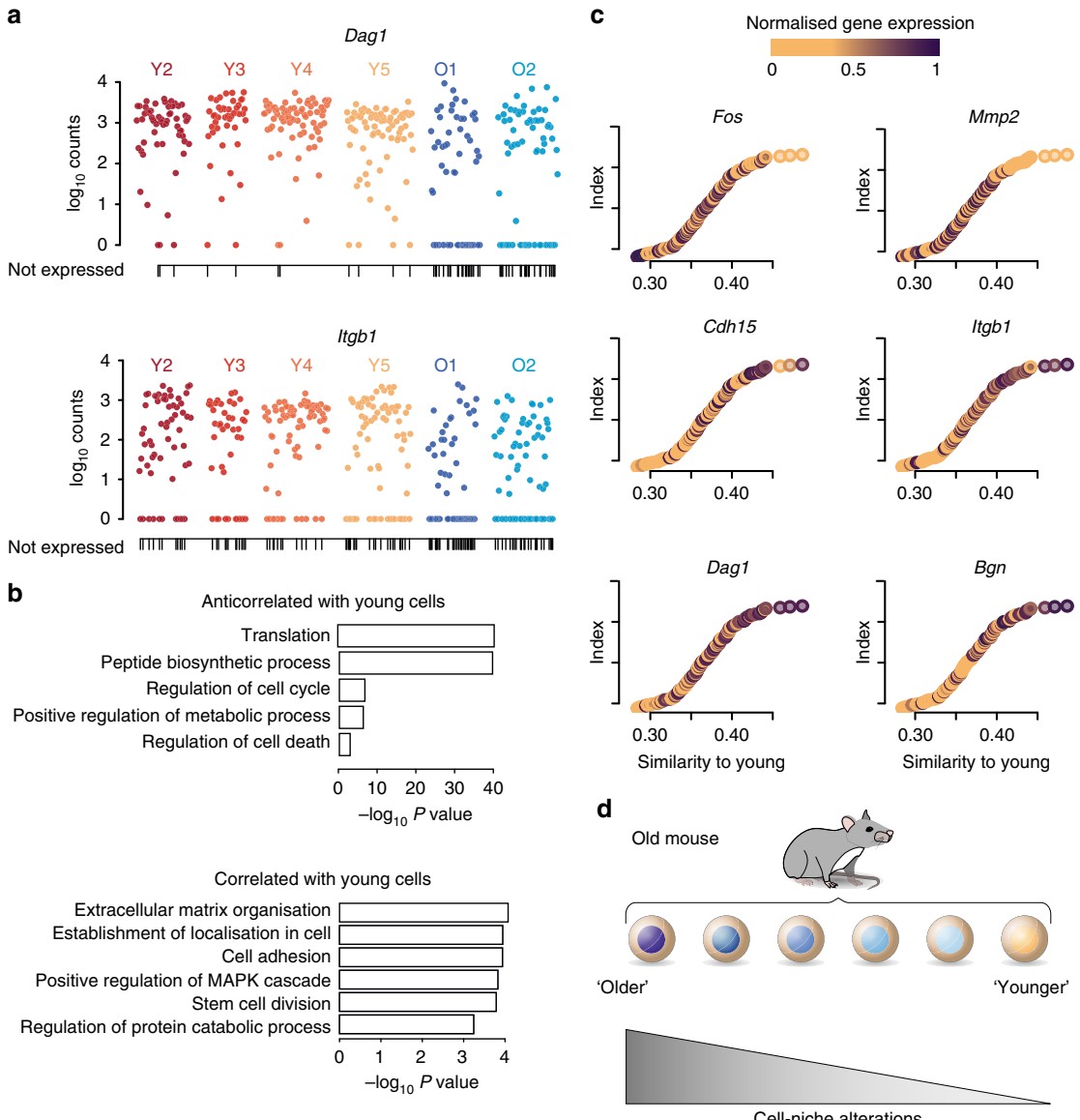

**Fig. 2** Variability within aged muscle stem cells and cell-niche interactions. **a** *Dag1* and *Itgb1* expression in young and old cells. Each dot represents a cell. Vertical lines on the *x*-axis indicate cells that do not express the gene. Expression frequencies, *Dag1*: young 0.96, old 0.65; *Itgb1*: young 0.79, old 0.59. **b** *P*-values of the GO terms associated with the top 200 anticorrelated (top) and correlated (bottom) genes with the similarity score to young cells. **c** Similarity between old and young cells. Each dot represents an old cell; on the *x*-axis old cells are ordered according to their similarity (Spearman correlation coefficient) to young cells (young mean expression). Colours indicate the normalised levels of expression of selected genes correlated across ordered cells. Source data are provided in Supplementary Data 3. **d** Model: aged muscle stem cells that have the most different transcriptomes from young stem cells are likely to have impaired stem cell-niche interactions and are more prone to exit quiescence

young individuals, a significant fraction (10–15%) showed low to no expression in old individuals (Supplementary Fig. 8). Overall, these observations confirm the scM&T-seq data and show increased expression variability with age at the protein level.

The observed increase in transcriptional variability with age could reflect the presence of cell subpopulations, or be a purely stochastic process. Despite not observing clear substructure (Fig. 1b, d right), we further investigated the origin of this variability by ranking old cells based on their transcriptome-wide similarity to young cells, and performed correlation analyses to identify the genes driving this ranking. Gene ontology analysis indicated that old cells that differed the most from young cells were enriched in processes such as translation and peptide biosynthesis (Fig. 2b top), while old cells that were most similar to young ones had higher levels of extracellular matrix-related

functions compared to the old cells that differed with the young (Fig. 2b bottom). For example, *Fos* and *Mmp2* were preferentially expressed in the most different old cells, while extracellular markers such as *Dag1*, *Itgb1*, *Cdh15* or *Bgn* were expressed in the most similar ones (Fig. 2c). These results indicate that cells that have accumulated more differences with age are likely to have impaired cell-niche interactions and are more prone to exit quiescence (Fig. 2d).

**DNA methylation-based age prediction.** Epigenetic age has so far not been assayed in adult stem cells, presumably because of the difficulties in obtaining sufficient material and isolating specific cell types. Taking advantage of our specific cell population, we next studied how DNA methylation-based age predictors[25]

behave in adult stem cells. Due to the sparse coverage of single cell data (average CpG sites covered per cell = 1.6 million), we aggregated cells by individual (two young and two old samples, 35 cells per individual) and built a model using 140 training samples using the same methods as previously described[25] (see "Methods"). The median absolute error in the training set was 0.3 weeks, and 3.2 weeks in the test set. These prediction accuracies are in line with the original report[25]. The model is based on 474 CpG sites that are covered in all training samples and the aggregated single cells at a 5X depth. The median absolute error of four bulk muscle samples that were left out completely from the modelling was 2.2 weeks, demonstrating that our model is able to predict accurately the age of the relevant tissue. Surprisingly, the predicted age for muscle stem cells was similar for all our samples: 9.2 and 9.3 weeks in the young samples and 10.2 and 11.6 weeks in the old ones (Fig. 3a). In order to investigate the potential variability masked in our aggregated samples and to get more insight into the prediction error, we estimated the epigenetic age using different combinations of single cells for each sample. We performed 100 permutations by randomly removing 5% of the cells and calculated the epigenetic age in each subsample. Given the coverage of the single cells, we used only the permutations with at least 4X coverage of each of the clock sites, resulting in between 36 and 84 permutations per sample. This analysis showed a significant increase in the epigenetic age of old samples; however, the predicted ages were still notably different from the actual chronological ages (Fig. 3a).

**DNA methylation changes and epigenetic drift**. We then investigated age-associated DNA methylation changes at the single cell level in greater detail. We discarded cells with <1 million paired-end alignments or <500,000 CpG and limited potential biases due to uneven sequencing depth between cells or different number of cells per individual by randomly subsampling 1 million reads from each cell and 35 cells per individual. We then selected two young and two old samples for the downstream analysis (2 million CpG sites on average per cell, "Methods", Supplementary Table 1). Global mean DNA methylation levels were around 50%, as previously reported for human primary cultured myoblasts[26] (Supplementary Fig. 9C). As with the transcriptomes, we did not observe clear subpopulations in any of the methylome samples (Supplementary Fig. 10). Overall, CpG islands, promoters and enhancers were hypomethylated (average 0.03%, 0.9%, 0.6% respectively); exons, myoblast enhancers (marked by H3K27ac) and shores (flanking regions of the CpG islands) were around 30% methylated, while repeats and bodies of active genes (marked by H3K36me3) were highly methylated (~70%) (Fig. 3b). We found that DNA methylation levels increased slightly with age, as reported for human primary myoblast[19], mostly in repeat elements and H3K36me3 regions (Fig. 3c–e).

Identical average methylation levels for a given genomic region may reflect different scenarios, from uniform populations to completely random heterogeneous patterns (Fig. 3f). Since we did not observe substructure in our data (Supplementary Fig. 10), and as stochastic epigenetic drift has been suggested to be a major hallmark of ageing[4], we computed a score to measure levels of stochastic intrapopulation heterogeneity (Supplementary Fig. 11, "Methods"). As expected, our initial measure of heterogeneity depended on average methylation levels (Fig. 3g). Hence, we developed an independent measure of heterogeneity by calculating the distance between the observed heterogeneity for each genomic region and a rolling median (Fig. 3g, "Methods"). Interestingly, this analysis showed that different genomic contexts displayed different levels of methylation heterogeneity between

cells; for example, CpG islands were more heterogeneous than enhancers (Fig. 3h).

Global levels of methylation heterogeneity were similar between ages (Supplementary Fig. 12); we next computed localised Z-score comparisons between young and old to examine changes in specific genomic elements. Notably, methylation of LINE-1 elements became more homogeneous with age whereas regions marked by H3K27me3 became more heterogeneous (Fig. 4a). Specifically, LINE-1 elements also experienced the highest increase in absolute DNA methylation levels, both of which may reflect a coordinated mechanism to prevent deleterious somatic retrotranspositions during ageing. Most of the H3K27me3 regions were associated with genes that are repressed, but poised for rapid activation[27]. We hypothesise that this increase in methylation heterogeneity may contribute to an impaired transcriptional response upon activation.

Interestingly, we observed a negative correlation between changes in methylation levels and changes in methylation heterogeneity (Promoters: Pearson's coefficient = −0.35, $P <$ 2.2e-16, Fig. 4b). Regions becoming more homogeneous showed an increase in methylation, suggesting that de novo methylation enzymes (Dnmt3a,b) are recruited to specific sites and add methylation in a coordinated manner between cells. In contrast, regions becoming more heterogeneous showed a decrease in their methylation levels. The key DNA methylation enzymes (DNMTs and TETs) are either expressed at very low levels, or they show no clear differences in expression between ages (Supplementary Fig. 13). This suggests that age-associated epigenetic changes are not likely to be driven by alterations of epigenetic modifiers at the transcriptional level (Supplementary Fig. 13), although we cannot exclude the possibility that they are driven by changes in protein levels or enzymatic activities. In addition, DNA methylation turnover has been shown to occur during the exit of pluripotency when DNMT3s and TETs are co-expressed[28]. We hypothesise that DNA methylation turnover could also take place in the quiescent state (e.g. through DNMT3s and TETs) and that age-associated epigenetic alterations could result from errors during this turnover. Alternatively, despite the low-proliferative history of these cells, we cannot exclude the possibility that Pax7[Hi] cells accumulate DNA methylation changes during occasional cell cycle entry.

Epigenetic changes may contribute to the age-associated pattern of transcriptional heterogeneity. Although our approach does not directly support causality, it provides a starting point to investigate the molecular links between epigenetic heterogeneity and transcriptional changes. We explored this possibility by analysing the association between promoter DNA methylation and gene expression. We calculated a correlation coefficient for each cell and confirmed the expected negative correlation for methylation and transcription (Fig. 4c). Interestingly, old cells that were most transcriptionally different from young cells showed lower levels of correlation (Mann–Whitney–Wilcoxon test; $P < 0.05$, Fig. 4c). Furthermore, we calculated changes in transcriptional variability between young and old cells (see "Methods") and observed that promoters with increased methylation heterogeneity tended to have increased transcriptional heterogeneity (Mann–Whitney–Wilcoxon test; $P < 0.001$) (Fig. 4d). It appears therefore that deterioration of transcriptional coherence during ageing is associated with increased promoter methylation heterogeneity and with decreased connectivity between the epigenome and the transcriptome.

## Discussion
In summary, we report transcriptional and epigenetic signatures associated with ageing in a deeply quiescent population of muscle

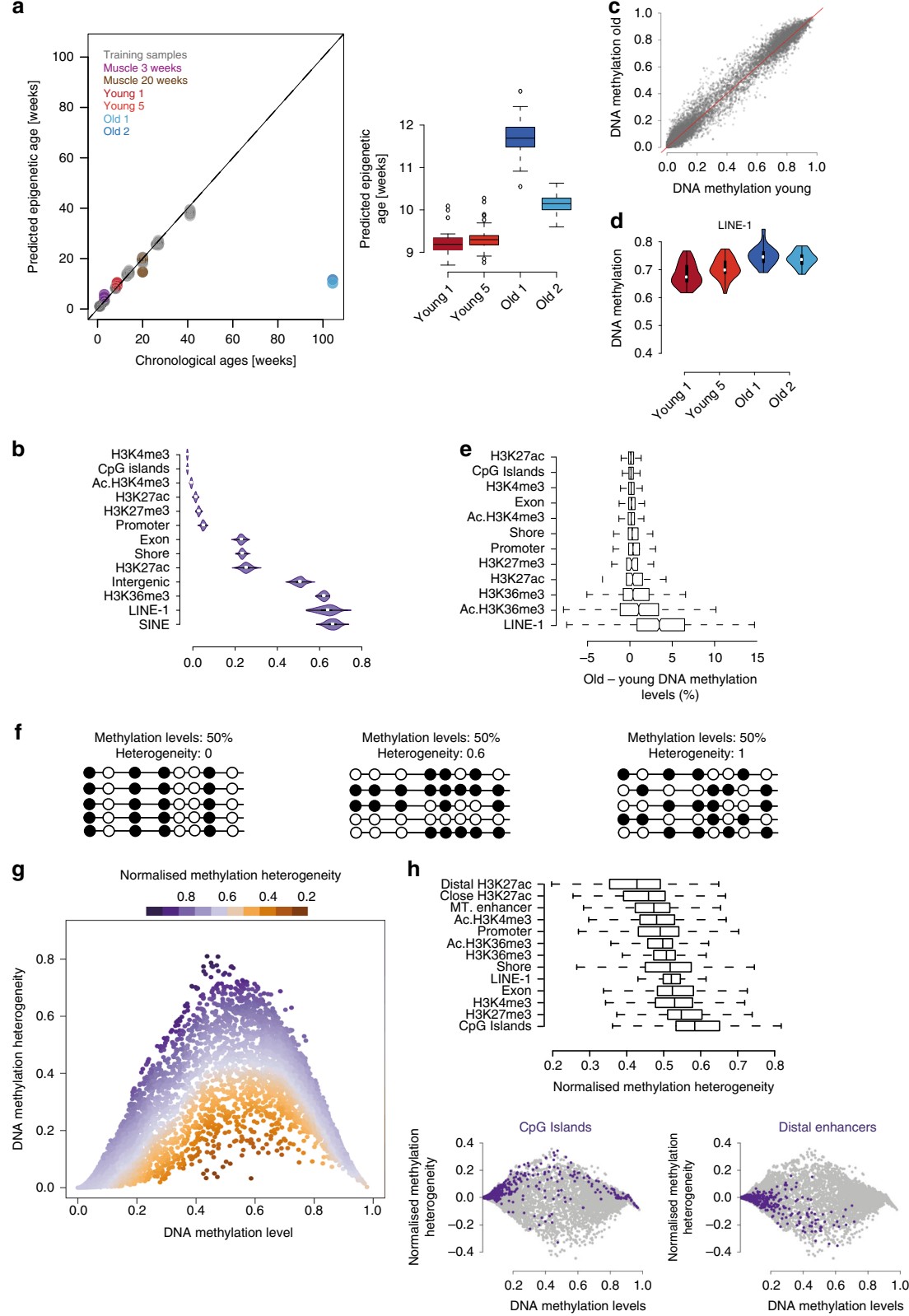

stem cells. While old stem cells show a slight increase in their DNA methylation age, this lags far behind their chronological age. DNA methylation-based age predictors are built with data from bulk tissues that represent a mix of different cell types. It is plausible that small changes in cell composition during ageing could affect the epigenetic age of a tissue[2]. Stem cells, which generally decrease in number during ageing might be one of these

cell populations. Consequently, the epigenetic clock might be a measure of the different proportions of stem and differentiated cells in a tissue. In accordance with this notion, several studies have shown that the CpG sites used in epigenetic clock models are enriched in polycomb target sites and are associated with developmental and cell differentiation genes[3,25,29]. Nonetheless, the slight increase in epigenetic age we observed in old stem cells

**Fig. 3** Changes in methylation levels and methylation heterogeneity. **a** Prediction of chronological age from a DNA methylation clock[25]. Left: x-axis shows the chronological age and y-axis shows the predicted epigenetic age. Grey circles indicate the training data sets. Test samples are represented in different colours: Muscle samples from Reizel et al.[42] are represented in purple (3 weeks-old) and brown (20 weeks-old). Pseudo bulk muscle stem cells from young animals (6–7 weeks-old) in red and from old animals (115 weeks-old) in blue. Right: Predicted epigenetic age of different combinations of single cells, permutations = 100. For all boxplots, the box represents the interquartile range and the horizontal line in the box represents the median (n = 100). **b** Levels of DNA methylation per cell across different genomic regions (Chip-seq data from 2-months-old mice[20,27], Ac: activated muscle stem cells). Circles inside the violin plots represent the median of the data and the boxes indicate the interquartile range. **c** Genome-wide mean methylation values in old and young cells. Each dot represents a genomic region. **d** Increase of DNA methylation with age across LINE-1 elements ($P < 0.05$). DNA methylation values computed per cell and individual. **e** Mean methylation difference between old and young cells across different genomic elements. For all boxplots, the box represents the interquartile range and the horizontal line in the box represents the median. **f** Examples of different distributions of DNA methylation heterogeneity at loci with similar average methylation. Empty circles represent unmethylated CpG sites and filled circles methylated CpG sites. **g** DNA methylation levels and DNA methylation heterogeneity. Each dot represents a genomic region from young or old cells. Colour scale represents the methylation-level normalised measure of DNA methylation heterogeneity. **h** Boxplot showing the normalised DNA methylation heterogeneity across different genomic elements in young cells (top). Normalised methylation heterogeneity and methylation levels across all the different genomic elements (grey) and across CpG Islands (purple) or enhancer regions (purple) in young cells (bottom). For all boxplots, the box represents the interquartile range and the horizontal line in the box represents the median

also suggests that cell-intrinsic changes are also likely to play a role. These changes could be related to cell metabolism or cell division. In this scenario, extrinsic and intrinsic factors would contribute to the epigenetic clock. Technical factors, such as the datasets used to train the model, could also contribute to these predictions. Future single cell studies including samples from different ages will be crucial to determine the specific role of cellular composition and cell-intrinsic changes in the epigenetic clock.

Previous studies have investigated transcriptional heterogeneity changes with age in mixed cell populations[30], which are affected by differences in cellular composition, such as an increase in senescent cells[30]. In contrast, our study is focused on a specific population of cells in which known stemness, activation and senescent markers were not affected by ageing. Even in this restricted population, we observed a global increase of uncoordinated transcriptional variability with age, indicating an intrinsic mechanism of cellular ageing. Interestingly, mouse muscle stem cells were shown to maintain clonal diversity during homoeostatic ageing by lineage-tracing[14], however, our study uncovers a dramatic underlying molecular heterogeneity in these stem cells that extends beyond maintenance of clonal homogeneity. We also observe that cells that have acquired more differences with age showed alterations in multiple extracellular matrix-related genes potentially affecting stem cell-niche interactions.

Elevated transcriptional variability with age has been reported in several studies[30–32], however the underlying causes remain largely unknown. The accumulation of somatic mutations only partially accounts for the increased cell-to-cell transcriptional variability[30], suggesting that epigenetic mechanisms might be a contributing factor[33]. In this study, by applying a combined single cell method for DNA methylation and the transcriptome, we show that epigenetic drift, or the uncoordinated accumulation of methylation changes in promoters, is associated with the increased transcriptional variability with age (Fig. 4e). Due to the deep quiescent state of the homoeostatic cells chosen for this study, our data highlight the possibility that the observed epigenetic patterns could be independent of extensive cell proliferation. We propose that this variability is detrimental due to uncoordinated transcription, thereby affecting the ability of stem cells to maintain quiescence, or activate coherently upon injury.

Understanding how molecular variability of stem cells affects their proliferation and differentiation capacities during aging will be of critical importance. In the present study, we restricted most of our analysis to cells expressing high levels of Pax7 to limit confounding factors, nonetheless, this might potentially co-select for other features. We interpreted the increase of variability in the

absence of clear substructure as a stochastic process, although we cannot formally exclude the possibility of the presence of as yet unidentified rare cell types or hidden population substructure. In future studies, the analysis of large numbers of cells from different stem cell populations and the integration of multiple molecular layers, will be highly informative for a more complete understanding of the underlying molecular mechanisms of stem cell ageing.

## Methods

**Mice.** Animals were handled according to national and European Community guidelines and an ethics committee of the Institut Pasteur (CETEA) in France approved protocols. Young (1.5–2.1 months) and old (23.3–27 months-old) *Tg: Pax7-nGFP*[17] mice, on a C57BL/6;DBA2 F1/JRj genetic background, were used in this study. Male littermates were euthanised around 9 am. Males were chosen preferentially to avoid interindividual variations due to asynchrony in the female oestrus cycle (Supplementary Table 1).

**Isolation of muscle stem cells.** Mice were sacrificed by cervical dislocation. *Tibialis anterior* muscles were dissected and placed into cold DMEM (Thermo-Fisher, 31966). Muscles were then chopped and put into a 15 ml Falcon tube containing 10 ml of DMEM, 0.08% collagenase D (Sigma, 11 088 882 001), 0.1% trypsin (ThermoFisher, 15090), 10 μg/ml DNaseI (Sigma, 11284932) at 37 °C under gentle agitation for 25 min. Digests were allowed to stand for 5 min at room temperature and the supernatants were collected in 5 ml of foetal bovine serum (FBS; Gibco) on ice. The digestion was repeated three times until complete digestion of the muscle. The supernatants were filtered through a 70-μm cell strainer (Miltenyi, 130–098–462). Cells were spun for 15 min at 515 × g at 4 °C and the pellets were resuspended in 1 ml freezing medium (10% DMSO (Sigma, D2438) in foetal calf serum (FCS, Invitrogen)) for long term storage in liquid nitrogen.

Before isolation by FACS, samples were thawed in 50 ml of cold DMEM, spun for 15 min at 515 × g at 4 °C. Pellets were resuspended in 300 μl of DMEM 2% FCS 1 μg/mL propidium iodide (Calbiochem, 537060) and filtered through a 40-μm cell strainer (BD Falcon, 352235). Viable cells were isolated based on size, granulosity and GFP expression level (total nGFP+ cells or top 10% nGFP^Hi cells, Supplementary Fig. 14) using a MoFlo Astrios cell sorter (Beckmann Coulter).

Single cells from the same sample were collected in 2.5 μL cold RLT Plus buffer (Qiagen, 1053393) containing 1U/μL RNAse inhibitor (Ambion, AM2694) and sorted in a 96 well-plate (LoBind Eppendorf, 0030129504), flash-frozen on dry ice and stored at −80 °C.

**BrdU administration and immunostaining.** Six days prior to harvesting, mice were given the thymidine analogue 5-Bromo-2'-deoxyuridine (BrdU, 1 mg/ml, Sigma, B5002) in the drinking water supplemented with sucrose (25 mg/ml). Muscle stem cells were isolated by FACS, plated on μ-Slide 8 Well (Ibidi, 80826) in medium (40% DMEM, 40% MCDB (Sigma), 20% FBS, 2% Ultroser (Pall), 1% Penicillin-Streptomycin (ThermoFisher)) at 37 °C 5% $CO_2$ 3% $O_2$ to allow adherence, fixed in 4% paraformaldehyde (PFA, Electron Microscopy Sciences) for 5 min at RT, washed twice 5 min with PBS, permeabilised in 0.5% Triton X-100 (Sigma) for 5 min at RT, washed three times with PBS, incubated with DNAse I (Roche, 04536282001) at 1 unit/μL in PBS at 37 °C for 30 min, washed three times with PBS, blocked in 10% goat serum (GS, Gibco) for 30 min at RT, incubated with mouse anti-BrdU antibody (1/100, BD, 347580) in 2% GS for 2 h at RT, washed three times 5 min in PBS, incubated with anti-mouse Cy3 secondary antibody

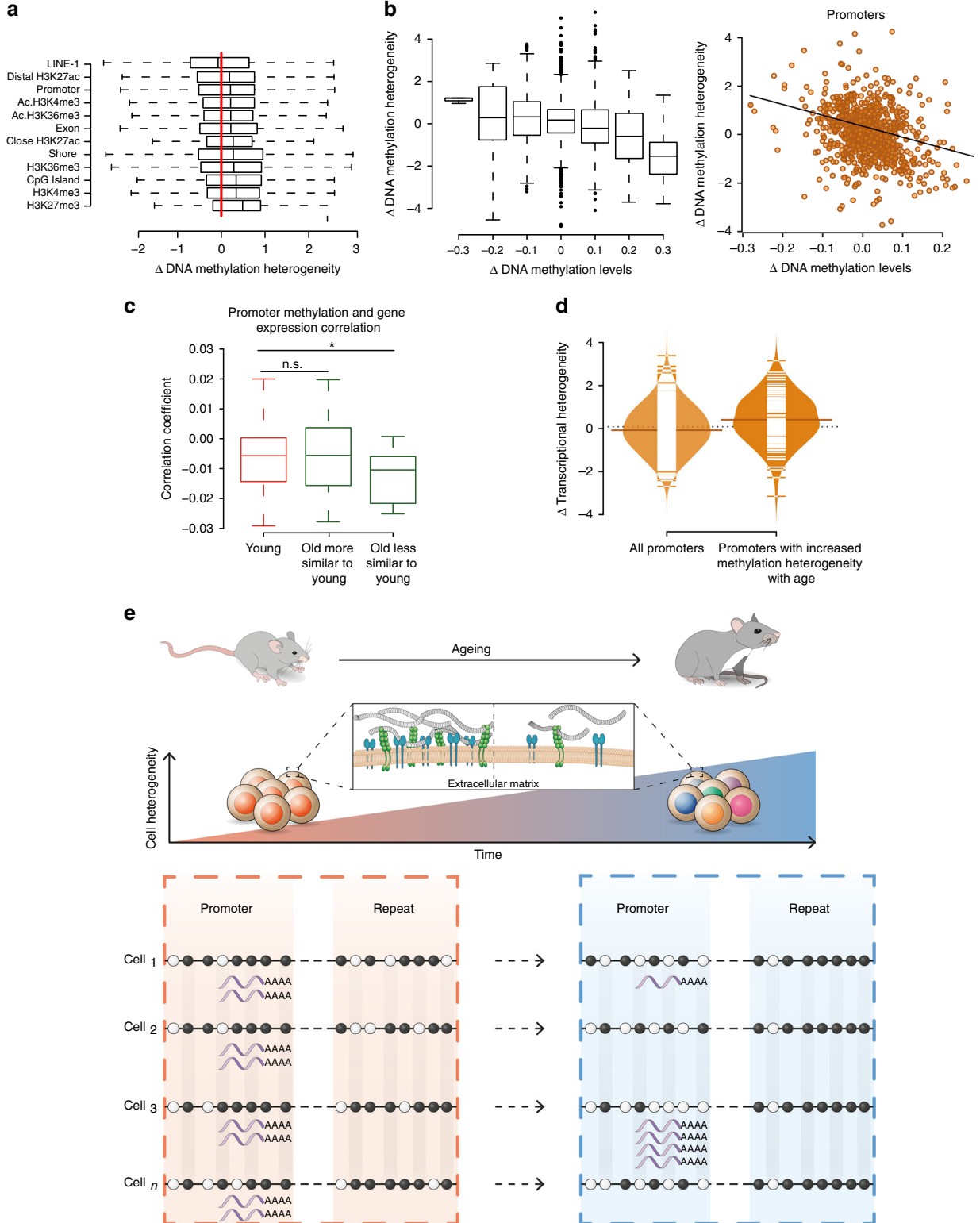

(1/500, Jackson ImmunoResearch, 115-165-205) and Hoechst (1 µg/ml) in 2% GS for 45 min at RT, and washed four times with PBS. Images were taken with a Zeiss LSM800 confocal (×40 objective).

**Single myofibre isolation and immunostaining.** Single myofibres were isolated from *Extensor digitorum longus* (EDL) muscles as previously described[34]. EDL muscles were dissected and incubated in 0.1% w/v collagenase (Sigma, C0130) in DMEM for 1 h in a 37 °C shaking water bath at 40 rpm. Following enzymatic digestion, mechanical dissociation was performed to release individual myofibres that were then fixed in 4% paraformaldehyde (PFA, Electron Microscopy

Sciences) for 5 min at RT, washed three times 10 min with PBS, permeabilised in 0.5% Triton X-100 (Sigma) for 5 min at RT, washed three times 10 min with PBS, blocked in 10% normal donkey serum (DS, Abcam, ab7475) for 1 h at RT, incubated with chicken anti-GFP antibody (1/1000, Abcam, ab13970) and mouse anti-M-Cadherin antibody (1/50, Nanotools, clone 12G4) or goat anti-ItgB1 antibody (1/100, SantaCruz, sc-9936) in 2% DS overnight at 4 °C, washed four times 10 min in PBS at RT, incubated with anti-chicken AlexaFluor 488 (1/500, ThermoFisher, A-11039) and anti-mouse Cy3 antibody (1/500, Jackson ImmunoResearch, 115-165-205) or anti-goat DyLight 550 (1/500, Diagomics, DKXGT-003) and Hoechst (1 µg/ml) in 2% DS for 45 min at RT, washed four times 10 min with PBS and mounted in PBS/Glycerol 75%. Images were taken

**Fig. 4** Changes in cell-to-cell methylation heterogeneity during ageing. **a** Normalised methylation heterogeneity changes with age (Δ methylation heterogeneity: old–young) across different genomic features (Chip-seq data from 2-months-old mice[20,27], Ac: activated muscle stem cells). For all boxplots, the box represents the interquartile range and the horizontal line in the box represents the median. **b** Genome-wide normalised methylation heterogeneity difference with ages (Δ methylation heterogeneity: old–young) binned by 0.1 methylation level differences (left). Changes in promoter methylation heterogeneity (y-axis) and methylation levels (x-axis) with age (right). For all boxplots, the box represents the interquartile range and the horizontal line in the box represents the median. Source data are provided in Supplementary Data 4 ($n = 2,355$). **c** Distribution of Pearson's correlation coefficients between promoter DNA methylation and gene expression (one association test per cell, number of cells: young = 64, old more similar to young = 30, old less similar to young = 20, *$P < 0.05$). For all boxplots, the box represents the interquartile range and the horizontal line in the box represents the median. **d** Increase of transcriptional heterogeneity with age across all promoters ($n = 394$) and promoters with increased DNA methylation heterogeneity (Δ methylation heterogeneity > 0.3, $n = 113$) ($P < 0.001$). Source data are provided in Supplementary Data 4. **e** Global increase of transcriptional cell-to-cell variability with age with enhanced heterogeneity in the multiple extracellular matrix-related genes (top). Relationship between transcriptional and DNA methylation heterogeneity in aged muscle stem cells (bottom). Empty circles represent unmethylated CpG sites and filled circles methylated CpG sites. Repeat elements become more homogeneous with age by increasing their methylation levels in a coordinated manner. In contrast, promoter regions become more heterogeneous by randomly losing DNA methylation, and this is coupled with an increase in transcriptional variability of the genes they drive

with a Zeiss LSM800 confocal (×40 objective) and processed with Imaris 7.2.1 software (Bitplane).

**Library preparation and data alignment**. We prepared scM&T-seq libraries[6] by isolating mRNA on magnetic beads and separating it from the single-cell lysate as described[35]. Subsequent, reverse transcription and amplification were performed on the beads using Smartseq2[36] but with 25 PCR cycles. Lysates containing genomic DNA were then processed according to a published single-cell bisulfite sequencing protocol[37]. Briefly, single cell gDNA was purified using AMPure XP beads (Beckman) prior to bisulfite conversion and purification (Zymo EZ-Direct). First strand DNA was synthesised using Klenow exo- (Enzymatics) and a primer containing Illumina read 1 sequence followed by a random hexamer at the 3′ end. This step was repeated four additional times for pre-amplification. Following purification, second strands were synthesised similarly but using a primer containing Illumina read 2 sequence after which the final library was amplified and indeces introduced. Single-cell RNA-seq libraries were aligned using HiSat2 with options –sp 1000,1000–no-mixed–no-discordant[38]. Single-cell bisulfite libraries were processed using Bismark[39] as described[6]. Mapped RNA-seq data were quantitated using the RNA-seq quantitation pipeline in Seqmonk software (www.bioinformatics.babraham.ac.uk/projects/seqmonk/).

**Quality control RNA-seq**. Cells expressing fewer than 1000 genes or <$10^5$ mapped reads allocated to nuclear genes were removed in quality control (Supplementary Figs. 15, 16). These cells were also verified to have less than 10% of mapped on mitochondrial genes (Supplementary Figs. 17A, B, 18A). Out of the 768 cells that were captured across the experiment from four young and two old mice, 377 passed our quality and filtering criteria (Supplementary Table 1). All samples showed similar distribution of reads and numbers of detected genes (Supplementary Figs. 17C, D, 18C).

**Data analysis RNA-seq**. Gene expression levels were estimated in terms of reads per million of mapped reads to the transcriptome. A score of variability per gene (named distance to the median) was calculated by fitting the squared coefficient of variation as a function of the mean normalised counts and then calculating the distance to a rolling average (window size = 100, Supplementary Fig. 19)[21]. We included only genes with an average normalised read count of at least 10. The top 1000 most variable genes of the entire data set were used to perform principal component analyses (as log$_2$-transformed and median-cantered values) (Fig. 1b, Supplementary Data 2). Single cell differential expression (SCDE) was used to calculate differential expression analysis between young and old cells (Supplementary Data 1)[40]. Cell cycle prediction was performed using the Cyclone function within the scran R package[18].

Cell-to-cell correlation analyses were performed using the top 500 most variable genes within each individual and using Spearman's correlation as the measure of similarity between cells (Fig. 1d). Correlation analyses were also performed using cohorts of 10 cells from the same individual (1000 iterations, Supplementary Fig. 4). Distance to the median of the top 500 most variable genes within each individual was computed for Fig. 1e, similar results are observed when restricting the analysis to genes that are expressed in all the individuals (average normalised read count of at least 10) and different numbers of genes (Supplementary Fig. 20).

An average young reference transcriptome was computed by calculating the mean of log transformed expression values for each gene across cells from young individuals. We then performed Spearman's correlation analyses to assess the similarity between each cell from old samples and the young transcriptome. Spearman's correlation analyses were then also used to find gene expression patterns associated with this genome-wide similarity score. Genes expressed in fewer than five cells were excluded from the analysis. The top 200 correlated and

anticorrelated genes (Supplementary Data 3) were used for GO enrichment analysis[41].

**Modelling the epigenetic clock**. Epigenetic ages of four samples (two young and two old, Supplementary Table 1) were calculated using a method based on the clock published in Stubbs et al.[25]. To predict the ages on the four single cell samples we first aggregated all the reads per cell by summing the methylation calls. We then took the two largest datasets used in the original publication, the Stubbs et al. data[25] and the Reizel et al. data[42], intersected them with the requirement that sites are found in at least 70% of the samples of these joint sets. Subsequently sites that were not covered 5× in all the merged single cell samples were removed. Lastly, all training samples that did not have 100% coverage of the sites obtained by these intersections were removed. This left 70,060 sites in 144 samples of the combined Reizel et al. and Stubbs et al. datasets. We used the same modelling approach to build the clock as described in the Stubbs et al. paper, with one exception; the methylation was binned in values of 20%, based on rounding off the methylation calls to assure there is no read-depth information encoded in the methylation calls.

**DNA methylation heterogeneity**. Cells that had less than 1 million paired-end alignments or less than 500,000 CpG sites covered were discarded (Supplementary Fig. 9). To avoid biases that might occur due to different sequencing depths or number of cells between individuals, the data was down-sampled to 1 million reads for each cell and randomly selected 35 cells from each individual (two young and two old, Supplementary Table 1). ChIP-seq datasets for H3K4me3, H3K27me3, H3K36me3 in muscle stem cells and H3K27ac in myoblast were obtained from existing studies[20,27]. Bowtie2 and MACS2 were used for mapping and peak calling respectively.

We developed a heterogeneity score based on Hamming distances and Shannon entropy between cell pairs from the same sample. This value captures the properties we desire: (i) ability to detect cell-to-cell stochastic heterogeneity; (ii) not affected by population substructure; (iii) not biased by missing values. Specifically, let $r$ be a matrix with methylation values of cells for a particular gene, each row corresponding to a cell and each column corresponding to a CpG site, and $w$ be the weight corresponding to the number of covered CpGs within each pairs of cells. For each pair of cells ($c$), we then computed the Hamming distance ($D$) and the Shannon entropy score of the pairs ($S$) considering sites with coverage in both cells. Then weighted heterogeneity score of the regions is:

$$H(r) = \frac{\sum_{c=1}^{n} w_c \times D_c}{\sum_{c=1}^{n} w_c} \times \frac{\sum_{c=1}^{n} w_c \times S_c}{\sum_{c=1}^{n} w_c}$$

Here $D_c$ is the normalised Hamming distance of a given a pair of cells, which measures the number of bits that are different in two binary sets:

$$D_c = \sum_{i=1}^{k} |x_i - x_j|$$

$S_c$ is the joint Shannon Entropy between a pair of cells which measures the complexity of the pattern:

$$S_c = -\sum_{i=1}^{k} p_i \times \log_2(p_i)$$

Here $p$ is the frequency of pairs of methylation values.

We validated our approach by applying the method in simulated data with increasing levels of methylation heterogeneity (Supplementary Fig. 11). We also observed that our algorithm is highly robust to missing data (Supplementary Fig. 11).

We applied this method across multiple genomic regions for each individual independently and then computed the average of young and old samples. Pairwise comparisons with fewer than four CpG sites were not considered in the analysis. Furthermore, to avoid misinterpretations because of poor coverage depth we excluded regions with: (i) <20 CpG sites, (ii) less than an average of two CpG sites covered per cell, (iii) less than 100 cell-to-cell pairwise comparisons. We also excluded regions with high coverage differences between ages (more than an average of 10 CpG sites or more than 200 cell-to-cell pairwise comparisons). A total of 63,823 genomic regions were used in the analysis (average window size = 2267 bp).

Coverage-weighted cell methylation values were used to calculate the mean methylation levels of each region. A normalised measure of DNA methylation heterogeneity was calculated for each region (from young or old samples) by fitting the score of heterogeneity as a function of the mean methylation levels and then calculating the distance to a rolling median of 1000 observations (Fig. 3g). Regions with less than 0.05 or more than 0.9 mean methylation levels were excluded from the analysis.

Differences between young and old DNA methylation heterogeneity values were Z-score normalised using a sliding window of 100 observations ordered by the mean value of young and old (Supplementary Fig. 21 and Supplementary Data 4). Same approach was used to calculate differences between young and old transcriptional heterogeneity (mean distance to the median) (Supplementary Fig. 19 and Supplementary Data 4).

**Reporting summary**. Further information on research design is available in the Nature Research Reporting Summary linked to this article.

## Data availability

Sequencing data have been deposited in GEO with the accession GSE121364. All other relevant data supporting the key findings of this study are available within the article and its Supplementary Information files or from the corresponding author upon reasonable request. The source data underlying Supplementary Fig. 2 are provided as a Source Data file. A reporting summary for this Article is available as a Supplementary Information file.

## Code availability

Custom software is available upon request.

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

## Acknowledgements

We would like to thank the Flow Cytometry Platform of the Center for Technological Resources and Research (Institut Pasteur), the Wellcome Trust Sanger Institute sequencing facility for assistance with Illumina sequencing and SciArtWork for producing illustrations. We also thank Laura Benson for assistance with library preparation. This project was supported by grants from Institut Pasteur, Agence Nationale de la Recherche (Laboratoire d'Excellence Revive, Investissement d'Avenir; ANR-10-LABX-73), Association Française contre les Myopathies (21857), CNRS, the European Research Council (Advanced Research Grant 332893), and the Biotechnology and Biological Sciences Research Council (BBSRC, CBBS/E/B/000C0425). I.H.-H. was supported by a

Marie Sklodowska-Curie Individual Fellowship (751439). M.J.B. was supported by a fellowship from the EMBL Interdisciplinary Postdoc (EI3POD) program under Marie Skłodowska-Curie Actions COFUND (grant number 664726).

## Author contributions

I.H.-H., B.E., T.S., S.T. and W.R. proposed the concept and designed the experiments. B.E. and P.-H.C. performed FACS; T.S. and S.C. performed library preparation and sequencing. I.H.-H. developed the analysis methodologies and analysed the experiments with advice from S.A. M.J.B. analysed the epigenetic clock information. I.H.-H., B.E., S.T. and W.R. wrote the paper. S.T. and W.R. contributed equally to this work. All authors read and agreed on the manuscript.

## Additional information

**Competing interests:** W.R. is a consultant and shareholder of Cambridge Epigenetix. T.S. is CEO of Chronomics. All other authors declare no competing interests.

