## [Peer Review File · Nature Communications]

Reviewers' comments:

Reviewer #1 (Remarks to the Author):

My main concerns about the manuscript titled "Ageing affects DNA methylation drift and transcriptional cell-to-cell variability in muscle stem cells" are the following:

1. The paper is very concise and lacks some important details. For example, it is not clear how many young and old animals were tested in this experiment. I can only judge based on the figures but they also not consistent (Fig 1D and E; S1B).
2. One of the main conclusions is that "Epigenetic drift, by accumulation of stochastic DNA methylation changes in promoters, is a substantial driver of the degradation of coherent transcriptional networks" (taken from the Abstract). Since investigated muscle stem cells are mitotically inactive, where the epigenetic drift is coming from? Isn't it more likely that transcriptomic networks are the primary cause of instability and stochastic age-dependent changes, which may differ across different cells. And DNA methylation/demethylation mRNAs and enzymes, which are part of the transcriptomes and proteomes, have only secondary effect? In general, the mechanisms of DNA methylation changes/drift in mitotically inactive cells are not clear and should be discussed in greater detail. The authors attempted to add some information on the DNA methylation machinery but it is not clear what is shown in Fig S5. What is meant by "expression levels of ... the enzymes"? What are the "normalized counts" and do they really differ by 10 orders of magnitude??
3. In connection to the previous comment, is it really the case that Pax7 gene expression is a sufficient indicator of mitotic quiescence? Is it possible that ageing stem cells would become more mitotically active despite high Pax7 gene expression, and this would be the actual reason for the increased variation of DNA methylation in the ageing stem cells?
4. It would be very helpful, actually necessary, to show the degrees of technical variation. How technical variation compares to the biological differences within/between old and young cell samples? Fig 1D shows quite significant differences between the young cell samples, where one sample shows the highest correlation coefficients but two other young samples are in the middle toward the old samples. What would be the distribution of correlation coefficients if multiple cohorts of cells from the same animal are investigate separately (from the cell enrichment step) and then compared to each other?
5. I am not sure if the experimental design allowed the authors to differentiated transcriptional variability originating from the cell subpopulations vs purely stochastic process.
6. GO analyses do not provide much insight and can be omitted or shortened.
7. The study would have been much more interesting if the same experiment was performed on differentiated myocytes, and their transcriptomic and epigenomic changes compared to the ones

in the muscle stem cells. The authors noted that increased cell-to-cell variability has been detected in aging non-stem cells but data exist for either differentiated cells or stem cells but not both. It remains unclear to what extent ageing effects in differentiated cells are transmitted from the stem cells and to what extent such effects are accumulated during the differentiation .

Reviewer #2 (Remarks to the Author):

In this manuscript, Hernando-Herraez et al. investigate how transcriptomic and epigenetic variability change with age in mouse muscle stem cells. By employing scM&T-seq on FACS-sorted muscle satellite cells, the authors show that cell-to-cell transcriptional and methylation heterogeneity is increased in older mice compared to younger mice. Functionally, younger cells and older cells (with transcriptomes similar to younger cells) were enriched for genes involved in the extracellular matrix; older cells with divergent transcriptomes were more enriched for processes involved in translation, hinting at alterations in niche-specific interactions. Additionally, they show that DNA methylation increases slightly with age, primarily at active genes and among transposable elements (TEs), and that different genomic elements (such as promoters or CpG islands) have different levels of methylation heterogeneity. With age, methylation heterogeneity of most of these genomic elements increases, with the exception of TEs, whose heterogeneity decreases. Sites that had increased levels of methylation with age were found to decrease in methylation heterogeneity. Finally, the authors show that increased promoter methylation heterogeneity is correlated with increased transcriptional heterogeneity.

This article describes an interesting and invaluable resource, and a unique dataset at the single cell level, which will be of broad interest to the aging and epigenomics field. A couple of points need to be further clarified, and some complementary analyses need to be performed for reproducibility and statistical soundness.

Major comments:

1. The authors describe that old cells have notably more cells with dropouts than young cells. It would be important to check whether this has any relationship with library size (UMI/read counts) or PCR duplication per cell (which may be high with 25 cycles of PCR), which was not investigated, rather than just whether this has a link to gene expression levels.

2. For high dimensionality single cell data, PCA (Fig 1B) is not the best dimensionality reduction method to detect structure in the data, and t-SNE is the usual tool recommended. It would be important to include t-SNE analyses for the scRNA-seq and scMethylome, coupled with cluster detection methods (e.g. SC3, PAGODA) to confirm that the purified cells constitute an homogeneous population, and not an assortment of distinct cell states, which could explain the observed results. If multiple cell populations are detected, the downstream analyses need to be modified to account for this (repeated in each cell subcluster).

3. The authors mention the use of DNAm for the epigenetic clock described by Steve Horvath's work. A mouse compatible DNAm age clock has recently been published (PMID: 30348905). It would be important to see what can be said of the distribution of the DNAm age of each cell in the study, or if the coverage of each cell is insufficient, the age of the aggregated sample (each young and old animal).

4. Single cell profiling experiments are definitely costly and time consuming, so this reviewer understands why only 3 young vs. 2 old samples are analyzed. However, since variability between cells is the variable of interest, we feel that the authors need to discuss the potential sampling bias and how this may or may not hold with more animals.

5. The Pax7-nGFP model is very elegant and help purify cells of interest readily. However, it is important to note that this reporter gene is subject to the same variability with age than endogenous genes (i.e. since the GFP-locus is likely to also experience cell-to-cell variability in its transcription/methylation). Thus, I am concerned that in the old samples, a biased sample of cell may have been obtained by selecting based on high levels of GFP expression, which may or may not, by selecting for a specific subpopulation, extend to other loci. The authors need to address this caveat in the discussion.

Minor comments:

1. The authors need to specify the sex of the mice used, the genetic background for the Pax7-nGFP transgene, and the time of the day at which mice were euthanized (to account for circadian gene expression influence). Whether a single sex, isolated samples from each sex, or pooled sex samples were used needs to be explicated.

Reviewer #3 (Remarks to the Author):

In this manuscript, Hernando-Herraez et Al. have used single cell RNA seq combined with single cell bisulfite sequencing to examine epigenetic drift in aging satellite cells. Using satellite cells from young (2 months) or old (24 months) mice, transcriptome analysis revealed no differences in expression of myogenic regulatory factors Pax7, Myf5, and Myod1, but did show upregulation in Fosb, Egr1, and Mmp2 in the old satellite cell population. Further intra-population analysis revealed increased transcriptional heterogeneity in the aged satellite cell population, which the authors used to classify the old satellite cells into two classes – (i) those closely resembling the homogenous young satellite cells characterized by increased expression of genes relating to ECM functions, and (ii) those that differed more than the young satellite cells and exhibited increased peptide and protein biosynthesis genes. The authors further revealed that genes exhibiting the highest amount of age-related heterogeneity were collagen genes (Col4a2, Col5a3, Col4a1) and extracellular matrix related genes (Dag1, Sparc, Cdh15, Itgb1). Collectively, the authors propose that this age-related transcriptional heterogeneity likely results in reduced cell-niche interactions, causing satellite cells to exit quiescence.

Global DNA methylation analysis did not identify distinct subpopulations between the young and old satellite cell populations. To refine their results, the authors proceeded to intrapopulation DNA methylation analysis which revealed that different genomic regions expressed methylation heterogeneity consistent with age – specifically, LINE-1 elements became more homogenous with age whereas regions associated with H3K27me3 expressed increased heterogeneity with age. Because H3K27me3 regions are associated with genomic regions that are poised for rapid expression, increased methylation heterogeneity at the epigenetic level could be a likely factor resulting in the previously observed age-related increased transcriptional heterogeneity. Lastly, the authors revealed that increased promoter methylation heterogeneity arising from aging is correlated to increased transcriptional heterogeneity, effectively confirming that epigenetic drift results in increased transcriptional heterogeneity during satellite cell aging. Based on these results, they conclude that age related drift in DNA methylation leads to loss of a coherent transcriptional network.

Overall this is an very interesting manuscript that makes use of a combination of single cell technologies to provide strong evidence for the genetic drift in quiescent stem cells during aging. While the authors have already published the combination of single cell RNA-Seq and single cell methylation sequencing (scM&T-seq), this is the first application of the technique to understanding a biological problem. In particular, their finding that loci that lose methylation show greater heterogeneity of methylation is interesting as it suggests that this mechanism might drive the epigenetic drift. Elucidating the mechanism of the heterogeneity of methylation is beyond the scope of this manuscript but is likely to be of interest to many going forward. The manuscript looks very polished, and I have only minor comments:

1. The authors conclude in the abstract that the degradation of coherent transcriptional networks have a consequence on stem cell functional decline during aging. While it is agreed that the identified ECM-related genes are showing altered expression with aging, the relationship to functional decline is only inferred. This is fine to discuss in the main text, but should not be stated as

fact in the final sentence of the abstract. Please modify the final sentence of the abstract to remove the words "...with consequent stem cell functional decline..."

2. The methods section refers to papers rather than providing details on the single cell RNA seq conditions. It would be useful for the authors to provide a line or two of detail of how the barcoding is introduced with the beads to allow batch sequencing of the samples (also perhaps mentioning how transcripts from each mouse were distinguished).

Reviewer #4 (Remarks to the Author):

In this manuscript, Hernando-Herraez et al analyzed transcriptional heterogeneity together with alterations of DNA methylation with age by using single-cell transcriptome and DNA methylome profiling of mouse PAX7-high-expression muscle progenitor/stem cells. The authors suggested that ageing uncoordinatedly increases DNA methylation and transcriptional variability and the epigenetic drift might driver the degradation of coherent transcriptional network. Although the topic of this paper is interesting and the combination of single-cell RNA and DNA methylation sequencing is technically fancy, unfortunately, I found no conclusion in this paper could be sufficiently supported by evidences which were provided. In the whole manuscript, the authors did not provide even a single experimental validation, which made the only bioinformatics analysis quite superficial. The writing and organization of the figures was also quite sloppy. Several of my major concerns and suggestions are discussed as follows:

Main points:

1. One of the main weak points of the paper is none of the data presented in the paper was conclusive, or at least supported a causative logic reasoning. Instead of, I can only see very weak correlations or independent pieces of data. For instance, as a quite simple paper, the authors emphasize in the abstract and the last paragraph with one of the few conclusive sentences as "we show that epigenetic drift, or the uncoordinated accumulation of methylation changes in promoters, contributes to the increased transcriptional variability with age". However, in the manuscript, epigenetic drift and transcriptional variability, at best, was just two separated sections of data (with weak evidences. see my minor points below). The authors did not present even a single evidence to prove the causative link of them. To improve it, the authors should perform biological experiments, for instance, using genetic animal models and/or at least cell models to modify methylation and

check the transcriptional changes. As this is the key point of the paper, the suggested experimental validation is essential.

2. I was concerning about the quality of cell isolation and library preparation, thereafter data quality. As show in Tab S2, the cells that passed QC was very low. In some plates/individuals no cell passed QC, and this happened in both single-cell RNA and DNA methylation sequencing. Can authors explain it? Furthermore, I suggest the authors provide detailed information of the QC and data pre-process, such as mean and distribution of genes/reads, mitochondrial genes ratio of each single cell experiments.

3. I was also quite surprised that the authors only analyzed the PAX7-high expression cells. The power of the single-cell techniques is exactly to identify the cell heterogeneity of complex cell population. From the data that the authors presented, PAX7-high expression cells seem to be quite unique from cell clustering. The authors are therefore suggested to analyze all PAX7 positive cells, as it might significantly improve the impact of the paper. This is especially true, as the title of the paper reads "Ageing affects DNA methylation and transcriptional cell-to-cell variability in muscle stem cells".

4. In several sections, such as Fig. 1C, 2A, the authors only showed single genes as examples, the authors should analyze with GO or KEGG etc to check systemic pathway/function change, but instead of giving a few simple genes to suggest a pathway change.

Minor points:

1. The authors concluded, from figure1B, that no global differences with age and absence of sequencing-related batch effects. From my point of view, this data cannot reflect the batch bias, let alone the clustering of cells was based on two components of PCA analysis. Is there another result to support this point?

2. In figure1C, the gene expression of Myf5 which was claimed to be no difference between groups was actually appeared higher difference than those which were down and up regulated genes, such as Spry1 and Ccl19, with the same scale of axis. Is it a fallacy or other reasons? Therefore, again, pathway analysis is essential.

3. In the part of analysis of DNA methylation patterns, according to all results, I think the difference between old and young mouse does not make sense or is not significant, so the conclusion that methylation changes in promoters, contributes to the increased transcriptional variability with age may not be compelling.

4. Does the gene expression heterogeneity in line 87 and 88 (Col4a2, Col5a3, Col4a1, Dag1, Sparc, Cdh15 Itgb1) accordingly be able to see DNA methylation heterogeneity (Fig3 section)? And why only two genes were presented in Fig. 2A, where are the results of the other genes?

5. The writing of the manuscript was very sloppy. Such as, Fig S1 A and B were not cited in the main text. And several writing examples, such as "While several reports indicate a loss of clonal

diversity during early life stages little is known about how cell-to-cell variability at the molecular level is involved in stem cell ageing", a comma was missed between the word 'stages' and 'little'.

45 Reviewer #1 (Remarks to the Author):

My main concerns about the manuscript titled "Ageing affects DNA methylation drift and transcriptional cell-to-cell variability in muscle stem cells" are the following:

50 1. The paper is very concise and lacks some important details. For example, it is not clear how many young and old animals were tested in this experiment. I can only judge based on the figures but they also not consistent (Fig 1D and E; S1B).

55 We would like to thank the reviewer for thoroughly reading our manuscript, for their insightful comments, and for raising critical points. We processed eight samples for RNA-seq and six for bisulfite-sequencing. However, due to the small amount of starting material and low capture efficiency of single cell technologies, library preparation did not work for all our samples. After quality control, we included 4 young and 2 old individuals for the transcriptome analysis (samples labelled as: young 1, young 2, young 3, young 4, old 1 and old 2) and 2 young and 2 old individuals for the methylation analysis (samples labelled as: young 1, 60 young 5, old 1 and old 2). We have now clearly highlighted this information in the manuscript (line 72, line 130, line 147, line 261, line 323, line 355 and line 372). Furthermore, we have updated Table S2 (line 714) which summarizes sample information and number of cells that we used for the analysis.

65 2. One of the main conclusions is that "Epigenetic drift, by accumulation of stochastic DNA methylation changes in promoters, is a substantial driver of the degradation of coherent transcriptional networks" (taken from the Abstract). Since investigated muscle stem cells are mitotically inactive, where the epigenetic drift is coming from? Isn't it more likely that transcriptomic networks are the primary cause of instability and stochastic age-dependent changes, which may differ across different cells. And DNA methylation/demethylation mRNAs and enzymes, which are part of the transcriptomes and proteomes, have 70 only secondary effect? In general, the mechanisms of DNA methylation changes/drift in mitotically inactive cells are not clear and should be discussed in greater detail. The authors attempted to add some information on the DNA methylation machinery but it is not clear what is shown in Fig S5. What is meant by "expression levels of ... the enzymes"? What are the "normalized counts" and do they really differ by 10 orders of magnitude??

75 We thank the reviewer for raising this point. We agree that the current experiment does not allow us to determine causality and we have now discussed this limitation in the manuscript (lines 196-199, 242). Furthermore, we have carefully investigated the transcriptional changes in the key methylation and demethylation enzymes and clarified Figure S9 (previously Figure S5), which shows no clear differences in 80 the levels of the enzymes between ages (line 185). We also observe that Dnmt3a and Tet3 show no differences in variability with age either. Variability was difficult to address for the other methylation/demethylation enzymes due to their low expression levels. These results suggest that age-associated epigenetic changes are not likely to be driven by alterations at the transcriptional level in the quiescent state. Other factors such as protein levels or enzymatic activity during quiescence may underlie

85 the observed epigenetic changes. DNA methylation turnover has been shown to occur during the exit of pluripotency when DNMT3s and TET are co-expressed¹. We hypothesized that DNA methylation turnover could also take place in the quiescence state and that the age-associated epigenetic alterations could result from errors during this turnover. Alternatively, despite the low proliferative history of these cells, we cannot discard the possibility that Pax7^{hi} cells do still enter the cell cycle occasionally and accumulate DNA
90 methylation changes when replicating. We have now discussed all these hypotheses in the manuscript (lines 183-194).

3. In connection to the previous comment, is it really the case that Pax7 gene expression is a sufficient
95 indicator of mitotic quiescence? Is it possible that ageing stem cells would become more mitotically active despite high Pax7 gene expression, and this would be the actual reason for the increased variation of DNA methylation in the ageing stem cells?

We thank the reviewer for raising this point. In this study, we specifically investigated muscle stem cells
100 expressing high levels of *Pax7* since this is linked to our previous report that in young homeostatic muscles Pax7^{hi} cells were in a deep quiescent state, compared to Pax7^{Lo} cells, which are more poised for activation². Additionally, the mitotic inactivity of the Pax7^{hi} cells we investigated here during ageing has been inferred from this report and the paper from Chakkalakal *et al.* (Figure 1j)³, showing that during ageing, non-cycling muscle stem cells express higher levels of *Pax7* than cycling muscle stem cells. Following the reviewer's
105 comment, we have now investigated the quiescent state of satellite cells by measuring BrdU uptake *in vivo* in aged and young individuals (Figure S2). When considering the total Pax7-nGFP population, old satellite cells were cycling less frequently than young satellite cells (p=0.0045), although the differences were small. However, no significant difference was observed between young and old Pax7^{hi} cells. Therefore, we conclude that transcriptional and epigenetic variability is unlikely to be driven by differences in mitotic activity.

110 In addition, we applied a learning algorithm to our sc-RNA-seq data to assign a cell cycle stage to each cell⁴. As a result, we observed that (with the exception of one cell from sample Young 2) cells are unlikely to be cycling and show no differences between ages (Figure S1). We have included these analyses in the manuscript (Figure S1 and Figure S2) (lines 75-79, lines 286-300, lines 336-337).

115 4. It would be very helpful, actually necessary, to show the degrees of technical variation. How technical variation compares to the biological differences within/between old and young cell samples? Fig 1D shows quite significant differences between the young cell samples, where one sample shows the highest correlation coefficients but two other young samples are in the middle toward the old samples. What would
120 be the distribution of correlation coefficients if multiple cohorts of cells from the same animal are investigate separately (from the cell enrichment step) and then compared to each other?

We completely agree that technical variability is an important issue to address. Due to the low availability of starting material, experimental technical replicates are not feasible in single cell studies. In order to minimize
125 the effects of potential technical variability, we have used more than one individual per age and processed all samples in parallel. Furthermore, we observed no clear differences in sequencing depth (Figure S12 D) and

gene expression variability was calculated considering that technical variability affects mostly lowly expressed genes⁵ (Methods and Figure S13).

130 Following the reviewer's suggestion, we have now performed correlation analysis using cohorts of 10 cells
from the same individual. Results from 1,000 iterations show that cells from old individuals are clearly less
correlated than those from young, supporting the higher variability of older transcriptomes (Figure S3).
Furthermore, in these 1,000 iterations the mean correlation levels of young individuals were always higher
than the old ones ($P < 0.001$). We have now included these analyses in the manuscript (lines 93-96, lines
135 341-342).

5. I am not sure if the experimental design allowed the authors to differentiated transcriptional variability
originating from the cell subpopulations vs purely stochastic process.

140 We again thank the reviewer for the thoughtful comment. Although we interpreted the increase of variability
in the absence of clear substructure as a stochastic process, we agree that undetected rare cell types or
hidden population substructure may influence our findings. Following the reviewer's comment, we revised
the manuscript and toned-down our interpretations (lines 248-254).

145 6. GO analyses do not provide much insight and can be omitted or shortened.

We agree with the reviewer that GO analyses do not consider the gene expression values and may not be
highly informative. However, we think it can be useful for reader to have a broader overview and biological
feel of the results. Following the reviewer's comment, we have shortened this part in the current version of
150 the manuscript.

7. The study would have been much more interesting if the same experiment was performed on
differentiated myocytes, and their transcriptomic and epigenomic changes compared to the ones in the
muscle stem cells. The authors noted that increased cell-to-cell variability has been detected in aging non-
155 stem cells but data exist for either differentiated cells or stems cells but not both. It remains unclear to what
extent ageing effects in differentiated cells are transmitted from the stem cells and to what extent such
effects are accumulated during the differentiation.

We agree with the reviewer that investigating how the observed variability affects differentiation is an
160 interesting question to pursue. However, differentiated mononucleated myocytes are virtually absent from
adult homeostatic resting muscles. Mononucleated myocytes can be obtained *in vitro* following differentiation
of muscle stem cells, but transcriptome and epigenome data obtained from *in vitro* samples would be difficult
to compare to data obtained from *in vivo* muscle stem cells. Otherwise, following muscle injury, quiescent
muscle stem cells activate, give rise to proliferating committed progenitors that differentiate to post-mitotic
165 myocytes, which eventually fuse together or to damaged myofibers. While FACS strategies are well
established for isolating muscle stem cells or proliferating progenitors, there is currently no established
method for isolating pure mononucleated myocytes from regenerating muscles. In addition, several reports
indicate that a fraction of aged muscle stem cells enter senescence⁴ or apoptosis^{5,6} upon activation cues,

170 which accounts partially for the decreased regenerative capacity of aged muscles. We anticipate that
mononucleated myocytes present in aged regenerating muscles would derive from a subset of the quiescent
muscle stem cell population, that has retained 'young-like' proliferative and differentiation capacities. Aged
myocytes would therefore not represent homogeneously the initial muscle stem cell population and could in
this case be enriched in 'young-like' myocytes arising from 'young-like' muscle stem cells, that we already
175 identified in our analysis. Our approach allowed us to study variability while minimising key confounder
factors such as cell cycle or differentiation state. Nonetheless, it would be very interesting to develop a
method to overcome these limitations and pursue this type of work in future studies.

Reviewer #2 (Remarks to the Author):

180 In this manuscript, Hernando-Herraez et al. investigate how transcriptomic and epigenetic variability change
with age in mouse muscle stem cells. By employing scM&T-seq on FACS-sorted muscle satellite cells, the
authors show that cell-to-cell transcriptional and methylation heterogeneity is increased in older mice
compared to younger mice. Functionally, younger cells and older cells (with transcriptomes similar to
younger cells) were enriched for genes involved in the extracellular matrix; older cells with divergent
185 transcriptomes were more enriched for processes involved in translation, hinting at alterations in niche-
specific interactions. Additionally, they show that DNA methylation increases slightly with age, primarily at
active genes and among transposable elements (TEs), and that different genomic elements (such as
promoters or CpG islands) have different levels of methylation heterogeneity. With age, methylation
heterogeneity of most of these genomic elements increases, with the exception of TEs, whose heterogeneity
190 decreases. Sites that had increased levels of methylation with age were found to decrease in methylation
heterogeneity. Finally, the authors show that increased promoter methylation heterogeneity is correlated with
increased transcriptional heterogeneity.

This article describes an interesting and invaluable resource, and a unique dataset at the single cell level,
195 which will be of broad interest to the aging and epigenomics field. A couple of points need to be further
clarified, and some complementary analyses need to be performed for reproducibility and statistical
soundness.

200 We appreciate the reviewer's critical reading of the manuscript, concise summary of our findings and
recognition of the relevance of this study.

Major comments:

1. The authors describe that old cells have notably more cells with dropouts than young cells. It would be
important to check whether this has any relationship with library size (UMI/read counts) or PCR duplication
205 per cell (which may be high with 25 cycles of PCR), which was not investigated, rather than just whether this
has a link to gene expression levels.

We thank the reviewer for the suggestions. Unfortunately, the combined single cell methylation and
transcription method uses the SMART-seq2 protocol, which is a non-UMI-based method. Typically, PCR
210 duplicates can also be identified as reads that align to the same genomic coordinates using reference-based

alignment. However, in the SMART-seq2 protocol full length mRNAs are reverse transcribed, pre-amplified and then fragmented with a Tn5 transposase. Due to this fragmentation step, PCR duplicates that arise during the pre-amplification step cannot be identified by their mapping positions.

215 We tried to minimize the effects of potential technical variation by using more than one individual per
condition and processed all samples in parallel. Furthermore, we observed no clear differences in
sequencing depth between samples (Figure S12D) and gene expression variability was calculated
considering that technical variability affects mostly lowly expressed genes⁵ (Methods and Figure S13). We
also further extended the analysis of transcriptional variability and observed consistent higher variability in
220 old individuals when subsampling different cohorts of cells from the same individual (see answer to point 4 of
reviewer 1 and Figure S3) (lines 94-96, lines 341-342)

2. For high dimensionality single cell data, PCA (Fig 1B) is not the best dimensionality reduction method to
detect structure in the data, and t-SNE is the usual tool recommended. It would be important to include t-
225 SNE analyses for the scRNA-seq and scMethylome, coupled with cluster detection methods (e.g. SC3,
PAGODA) to confirm that the purified cells constitute an homogeneous population, and not an assortment of
distinct cell states, which could explain the observed results. If multiple cell populations are detected, the
downstream analyses need to be modified to account for this (repeated in each cell subcluster).

230 We thank the reviewer for the comment. Following the reviewer's comment, we have applied t-SNE on the
scRNA-seq and the sc-Methylome data exploring different perplexity values. The perplexity parameter can
be interpreted as measure of the number of neighbors for each point and it has a complex effect on the
results. In all the scenarios, we observe that t-SNE and PCA plots show similar results, indicating no clear
substructure of the data. Furthermore, by applying learning algorithm to our scRNA-seq data, we have
235 assigned a cell cycle stage to each cell and verified the dormancy state⁴. As a result, we observed that, with
the exception of one cell from sample Young 2, cells are unlikely to be cycling and show no differences
between ages (see response to point 3 of reviewer 1) (Figure S1). Therefore, cell cycle is unlikely to drive
potential cell substructure of our data. We also applied SC3 interrogating a range of number of clusters (*ks*
values=2:6). The separation of young and old individuals was identified as the best clustering (*ks*=2, average
240 silhouette with 0.91) identifying similar markers of the ones we report in Table S1. No other clear cell
populations were detected (average silhouette values < 0.6).

245 Figure #4: The two-dimensional outputs of t-SNE using the scRNA-seq data with different values of perplexity, between 5 and 50.

250 Figure #5 The two-dimensional outputs of t-SNE using single-cell mean gene body methylation data with different values of perplexity, between 5 and 50.

255 3. The authors mention the use of DNAm for the epigenetic clock described by Steve Horvath's work. A mouse compatible DNAm age clock has recently been published (PMID: 30348905). It would be important to see what can be said of the distribution of the DNAm age of each cell in the study, or if the coverage of each cell is insufficient, the age of the aggregated sample (each young and old animal).

260 We thank the reviewer for this interesting suggestion. Unfortunately, the aggregated coverage was
insufficient to test the published models. However, we built a new model based on the epigenetic clock
described in Stubbs *et al.*⁶ Detailed information is in the revised methods and results sections as well as in
Figure 3A. In short, we used elastic net modelling combined with a double loop cross-validation
described in Stubbs *et al.* on 140 of the training samples and we built a model with a median absolute error
of 0.3 weeks on the training set and a median absolute error of 3.2 weeks on the test set. These prediction
265 accuracies are in line with the original publication. The model is based on 474 CpG sites that are covered in
all training samples and muscle satellite cells with at least 5X coverage. We found that the epigenetic ages
of the old and young samples (pseudo bulk) are very similar, the two young samples have a predicted age of
9.2 weeks (Young 1), 9.3 weeks (Young 5), the old samples have a predicted age of 11.6 weeks (Old 1) and
10.2 weeks (Old 2) (Figure 3A). We observe a small difference between young and old muscle stem cells,
270 which is far smaller than the actual age difference and falls within the prediction error of the model (3.2
weeks). We therefore hypothesize that differentiation potential can be an important factor driving the
epigenetic clock. In this sense, the clock could be reflecting the different proportions of stem cells, that are
known to decrease with age. However, changes in post mitotic cells are also likely to play an important role.
Following the reviewer's suggestion, we have now included these analyses in the manuscript (Figure 3A)
275 (lines 30-31, line 39, lines 42-45, lines 126-141, lines 211-224, lines 354-366).

4. Single cell profiling experiments are definitely costly and time consuming, so this reviewer understands
why only 3 young vs. 2 old samples are analysed. However, since variability between cells is the variable of
interest, we feel that the authors need to discuss the potential sampling bias and how this may or may not
280 hold with more animals.

We thank the reviewer for this comment. We agree that a larger number of individuals would reduce potential
sample bias. Following the reviewer's comment, we have now discussed this issue in the manuscript.
Furthermore, to address whether cell sampling bias could affect our transcriptional variability measure, we
285 have performed correlation analysis using cohorts of 10 cells from the same individual (Figure S3). Results
from 1,000 iterations show that cells from old individuals are less correlated than those from young,
supporting the higher variability of older transcriptomes. Importantly, in these 1,000 iterations the mean
correlation levels of young individuals were always higher than the old ones ($P < 0.001$). We have now
included this analysis in the manuscript (Figure S3) (lines 94-96, lines 341-342, lines 252-254).

290 5. The Pax7-nGFP model is very elegant and help purify cells of interest readily. However, it is important to
note that this reporter gene is subject to the same variability with age than endogenous genes (i.e. since the
GFP-locus is likely to also experience cell-to-cell variability in its transcription/methylation). Thus, I am
concerned that in the old samples, a biased sample of cell may have been obtained by selecting based on
295 high levels of GFP expression, which may or may not, by selecting for a specific subpopulation, extend to
other loci. The authors need to address this caveat in the discussion.

We thank the reviewer for this comment. The selection of Pax7-nGFP^{Hi} cells allows us to investigate cellular
variability while minimising key confounder factors such as cell cycle or differentiation state. However, as the

300 reviewer indicates, we cannot dismiss the possibility that the selection of high levels of Pax7-nGFP may potentially co-select other features. We have discussed this caveat in the discussion (lines 248-251).

Minor comments:

305 1. The authors need to specify the sex of the mice used, the genetic background for the Pax7-nGFP transgene, and the time of the day at which mice were euthanized (to account for circadian gene expression influence). Whether a single sex, isolated samples from each sex, or pooled sex samples were used needs to be explicated.

310 We thank the reviewer for the comment. Male littermates were used, on a C57BL/6;DBA2 F1/JRj genetic background, euthanized around 9 am. Males were preferred to females to avoid interindividual variations due to asynchrony in the oestrus cycle. We have included that information in Table S2 and in the methods section of the manuscript (lines 260-264).

315

Reviewer #3 (Remarks to the Author):

In this manuscript, Hernando-Herraez et Al. have used single cell RNA seq combined with single cell bisulfite sequencing to examine epigenetic drift in aging satellite cells. Using satellite cells from young (2 months) or old (24 months) mice, transcriptome analysis revealed no differences in expression of myogenic regulatory factors Pax7, Myf5, and Myod1, but did show upregulation in Fosb, Egr1, and Mmp2 in the old satellite cell population. Further intra-population analysis revealed increased transcriptional heterogeneity in the aged satellite cell population, which the authors used to classify the old satellite cells into two classes – (i) those closely resembling the homogenous young satellite cells characterized by increased expression of genes relating to ECM functions, and (ii) those that differed more than the young satellite cells and exhibited increased peptide and protein biosynthesis genes. The authors further revealed that genes exhibiting the highest amount of age-related heterogeneity were collagen genes (Col4a2, Col5a3, Col4a1) and extracellular matrix related genes (Dag1, Sparc, Cdh15, Itgb1). Collectively, the authors propose that this age-related transcriptional heterogeneity likely results in reduced cell-niche interactions, causing satellite cells to exit quiescence.

330

Global DNA methylation analysis did not identify distinct subpopulations between the young and old satellite cell populations. To refine their results, the authors proceeded to intrapopulation DNA methylation analysis which revealed that different genomic regions expressed methylation heterogeneity consistent with age – specifically, LINE-1 elements became more homogenous with age whereas regions associated with H3K27me3 expressed increased heterogeneity with age. Because H3K27me3 regions are associated with genomic regions that are poised for rapid expression, increased methylation heterogeneity at the epigenetic level could be a likely factor resulting in the previously observed age-related increased transcriptional heterogeneity. Lastly, the authors revealed that increased promoter methylation heterogeneity arising from aging is correlated to increased transcriptional heterogeneity, effectively confirming that epigenetic drift results in increased transcriptional heterogeneity during satellite cell aging. Based on these results, they conclude that age related drift in DNA methylation leads to loss of a coherent transcriptional network.

340

Overall this is an very interesting manuscript that makes use of a combination of single cell technologies to provide strong evidence for the genetic drift in quiescent stem cells during aging. While the authors have already published the combination of single cell RNA-Seq and single cell methylation sequencing (scM&T-seq), this is the first application of the technique to understanding a biological problem. In particular, their finding that loci that lose methylation show greater heterogeneity of methylation is interesting as it suggests that this mechanism might drive the epigenetic drift. Elucidating the mechanism of the heterogeneity of methylation is beyond the scope of this manuscript but is likely to be of interest to many going forward. The manuscript looks very polished, and I have only minor comments:

We would like to thank the reviewer for thoroughly reading our manuscript, for their insightful comments, and for pointing out the importance and novelty of our work.

1. The authors conclude in the abstract that the degradation of coherent transcriptional networks have a consequence on stem cell functional decline during aging. While it is agreed that the identified ECM-related genes are showing altered expression with aging, the relationship to functional decline is only inferred. This is fine to discuss in the main text, but should not be stated as fact in the final sentence of the abstract. Please modify the final sentence of the abstract to remove the words "...with consequent stem cell functional decline..."

We thank the reviewer for the comment, we have now removed these words from the abstract.

2. The methods section refers to papers rather than providing details on the single cell RNA seq conditions. It would be useful for the authors to provide a line or two of detail of how the barcoding is introduced with the beads to allow batch sequencing of the samples (also perhaps mentioning how transcripts from each mouse were distinguished).

We thank the reviewer for this comment. Following the reviewer's comment, we have now expanded information about the library preparation in the methods section of the manuscript. We also would like to clarify that cells from different individuals were sorted in different plates. Therefore, we did not have to label the cells by sample. We have also clarified this information in the manuscript (line 282, lines 303-313)

Reviewer #4 (Remarks to the Author):

In this manuscript, Hernando-Herraez et al analyzed transcriptional heterogeneity together with alterations of DNA methylation with age by using single-cell transcriptome and DNA methylome profiling of mouse PAX7-high-expression muscle progenitor/stem cells. The authors suggested that ageing uncoordinatedly increases DNA methylation and transcriptional variability and the epigenetic drift might driver the degradation of coherent transcriptional network. Although the topic of this paper is interesting and the combination of single-cell RNA and DNA methylation sequencing is technically fancy, unfortunately, I found no conclusion in this paper could be sufficiently supported by evidences which were provided. In the whole manuscript, the

385 authors did not provide even a single experimental validation, which made the only bioinformatics analysis quite superficial. The writing and organization of the figures was also quite sloppy. Several of my major concerns and suggestions are discussed as follows:

We appreciate the Reviewer's critical reading of the manuscript, concise summary of our findings, recognizing the relevance of this study and for highlighting critical points.

390 Main points:

1. One of the main weak points of the paper is none of the data presented in the paper was conclusive, or at least supported a causative logic reasoning. Instead of, I can only see very weak correlations or independent pieces of data. For instance, as a quite simple paper, the authors emphasize in the abstract and the last paragraph with one of the few conclusive sentences as "we show that epigenetic drift, or the uncoordinated accumulation of methylation changes in promoters, contributes to the increased transcriptional variability with age". However, in the manuscript, epigenetic drift and transcriptional variability, at best, was just two separated sections of data (with weak evidences. see my minor points below). The authors did not present even a single evidence to prove the causative link of them. To improve it, the authors should perform biological experiments, for instance, using genetic animal models and/or at least cell models to modify methylation and check the transcriptional changes. As this is the key point of the paper, the suggested experimental validation is essential.

405 We thank the reviewer for the comment. We agree that experimental perturbations of epigenetic heterogeneity would ultimately provide the most compelling evidence. However, due to the nature of the changes, such validations are particularly challenging. Genetic knockouts and epigenetic editing do not allow the modification of methylation patterns in a non-coordinated manner, this would require extensive method development that is beyond the scope of this paper but will be of high interest for future studies if experimental models can be developed to test causality. Following the reviewer's comment, we revised the manuscript and toned-down our interpretations (lines 196-199, line 242).

2. I was concerning about the quality of cell isolation and library preparation, thereafter data quality. As show in Tab S2, the cells that passed QC was very low. In some plates/individuals no cell passed QC, and this happened in both single-cell RNA and DNA methylation sequencing. Can authors explain it? Furthermore, I suggest the authors provide detailed information of the QC and data pre-process, such as mean and distribution of genes/reads, mitochondrial genes ratio of each single cell experiments.

420 We thank the reviewer for the comment. We agree that QC is a critical step in single cell sequencing technologies. The small amount of starting material is an inherent limitation that usually leads to the presence of some low quality samples. This can be due to multiple factors that are difficult to control, including damage during the isolation procedure, RNA degradation or low capture efficiency. We first maximized the isolation of high-quality samples by including a viability marker at the time of FACS-isolation, keeping samples at 4°C during single-cell sorting and SNAP freezing sorted cells. Further, our pre-processing and quality control pipelines have previously shown to be effective in identifying and removing

425

potential low-quality cells and artefacts^{5,8,9}. For example, all cells have fewer than 10% of reads assigned to the mitochondrial genome and show similar distribution of reads and numbers of detected genes (Figure S12). Furthermore, we do not detect any outliers when applying different clustering methods (see response to point 2 of reviewer 2). Following the reviewer's comments, we have now included this information in the manuscript in the methods section and in Figure S12 (lines 322-325).

430

3. I was also quite surprised that the authors only analyzed the PAX7-high expression cells. The power of the single-cell techniques is exactly to identify the cell heterogeneity of complex cell population. From the data that the authors presented, PAX7-high expression cells seem to be quite unique from cell clustering. The authors are therefore suggested to analyze all PAX7 positive cells, as it might significantly improve the impact of the paper. This is especially true, as the title of the paper reads "Ageing affects DNA methylation and transcriptional cell-to-cell variability in muscle stem cells".

435

We thank the reviewer for the comment. We agree with the reviewer that investigating how variability affects the whole muscle stem cell population is a very interesting question. However, we focused on a defined population of stem cells to assess variability in a manner as accurate as possible, minimising key confounding factors such as cell cycle or differentiation states. We notably now measured the proliferation of young and old muscle stem cells in homeostatic muscles, through *in vivo* BrdU uptake (see response to point 3 of reviewer 1). Analysing the total Pax7-nGFP population, we observed that old satellite cells were cycling less frequently than young satellite cells ($p=0.0045$). However, no significant difference was observed between young and old Pax7^{Hi} cells (Figure S2, lines 75-79, lines 286-300). Therefore, we anticipate that comparing young vs. old total Pax7 cells would at least compare cells with different proliferative histories. Restricting our analysis to Pax7^{Hi} cells allows us to dismiss the possibility that the observed differences arise from different proliferation potential between ages. Beyond that we have now calculated that if we were to sequence the Pax7 total population, with $n=6$ mice (3 young + 3 old) and $n>250$ cells per mouse, this would mean eighteen 96-well plates of scRNA-seq and scBS-seq which would take approximately 9 months for library preparation, sequencing and analysis and would cost about £75k. This level of time and financial investment does not make sense to us for this manuscript and the conclusions it reaches.

440

445

450

4. In several sections, such as Fig. 1C, 2A, the authors only showed single genes as examples, the authors should analyze with GO or KEGG etc to check systemic pathway/function change, but instead of giving a few simple genes to suggest a pathway change.

455

We thank the reviewer for the comment. We have now systematically investigated potential pathway/function enrichments. The only category that is significant among the genes with increased transcriptional variability with age is "signalling receptor activity" (GO:0038023) (FDR q-value 2.37E-1).

460

Minor points:

1. The authors concluded, from figure1B, that no global differences with age and absence of sequencing-related batch effects. From my point of view, this data cannot reflect the batch bias, let alone the clustering of cells was based on two components of PCA analysis. Is there another result to support this point?

465

470 We thank the reviewer for this comment that also helped us clarify an important point. In order to minimize potential technical bias, we processed all samples in parallel. Cells were sorted into plates by individuals and individuals from the different ages were pooled into two sequencing lanes. Therefore, potential technical bias could be observed by individual or sequencing lane. The absence of clustering by individual in the PCA indicates absence of strong batch effects, which is also supported when clustering by t-SNE (Figure #9) and when interrogating different components of the PCA (Figure #10). Similar results are observed when considering sequencing lanes (Figure #11). Furthermore, all individuals show similar distributions of reads and number of genes (see point #2).

475

480 Figure #9: The two dimensional outputs of t-SNE using the scRNA-seq data with different values of perplexity, between 5 and 50.

485 Figure #10: Second and third (left) and third and fourth dimensional outputs of the PCA using the scRNA-seq data coloured by individual.

490 Figure #11: The two dimensional outputs of the PCA using the scRNA-seq data coloured by sequencing batch.

2. In figure1C, the gene expression of Myf5 which was claimed to be no difference between groups was actually appeared higher difference than those which were down and up regulated genes, such as Spry1 and Ccl19, with the same scale of axis. Is it a fallacy or other reasons? Therefore, again, pathway analysis is essential.

495 We thank the reviewer for pointing this out, we agree that Figure 1C could be misleading. In this plot we had represented the mean expression level of a given gene while the significance was assessed by SCDE¹⁰. This method is a specific probabilistic model for single cell data that takes more parameters into account than the mean to fit individual error models and assesses significance¹⁰. We have now plotted the log2 fold change and p-value to visualise differentially expressed genes (Figure 1C, line 555). This representation now shows more clearly that Myf5 shows no significant difference.

500 3. In the part of analysis of DNA methylation patterns, according to all results, I think the difference between old and young mouse does not make sense or not significant, so the conclusion that methylation changes in promoters, contributes to the increased transcriptional variability with age may not be compelling.

505 We thank the reviewer for this comment. Due to the lack of previous studies it is difficult to compare the magnitude of changes that we observed (Figure 4D: mean group 1: -0.0009, mean group 2: 0.42, P <0.001). We would like to highlight that the sparse coverage of processed single-cell epigenome data is one of the key limitations of the technology, which limits the number of genes with coverage on the transcriptome and the epigenome. Following the reviewer's comment, we revised the manuscript and toned-down our interpretations (lines 196-199, line 242).

510 4. Does the gene expression heterogeneity in line 87 and 88 (Col4a2, Col5a3, Col4a1, Dag1, Sparc, Cdh15 Itgb1) accordingly be able to see DNA methylation heterogeneity (Fig3 section)? And why only two genes were presented in Fig. 2A, where are the results of the other genes?

We thank the reviewer for this comment. Unfortunately, the coverage on these genes is not enough to assess their epigenetic heterogeneity. However, we observe an increase of epigenetic variability in other genes related to the extracellular matrix such as *Lamc3* (0.25 % increase in variability). Following the reviewer's comment, we have now added new examples in Figure S4.

525

5. The writing of the manuscript was very sloppy. Such as, Fig S1 A and B were not cited in the main text. And several writing examples, such as "While several reports indicate a loss of clonal diversity during early life stages little is known about how cell-to-cell variability at the molecular level is involved in stem cell ageing", a comma was missed between the word 'stages' and 'little'.

530

We thank the reviewer for identifying these mistakes. We have revised the writing extensively based on this comment.

535

References

1. Rulands, S. *et al.* Genome-Scale Oscillations in DNA Methylation during Exit from Pluripotency. *Cell Syst.* (2018). doi:10.1016/j.cels.2018.06.012
- 540 2. Rocheteau, P., Gayraud-Morel, B., Siegl-Cachedenier, I., Blasco, M. A. & Tajbakhsh, S. A subpopulation of adult skeletal muscle stem cells retains all template DNA strands after cell division. *Cell* **148**, 112–125 (2012).
3. Chakkalakal, J. V, Jones, K. M., Basson, M. A. & Brack, A. S. The aged niche disrupts muscle stem cell quiescence. *Nature* **490**, 355–360 (2012).
- 545 4. Scialdone, A. *et al.* Computational assignment of cell-cycle stage from single-cell transcriptome data. *Methods* (2015). doi:10.1016/j.ymeth.2015.06.021
5. Brennecke, P. *et al.* Accounting for technical noise in single-cell RNA-seq experiments. *Nat. Methods* (2013). doi:10.1038/nmeth.2645
6. Stubbs, T. M. *et al.* Multi-tissue DNA methylation age predictor in mouse. *Genome Biol.* (2017).
550 doi:10.1186/s13059-017-1203-5
7. Reizel, Y. *et al.* Gender-specific postnatal demethylation and establishment of epigenetic memory. *Genes Dev.* (2015). doi:10.1101/gad.259309.115
8. McCarthy, D. J., Campbell, K. R., Lun, A. T. L. & Wills, Q. F. Scater: Pre-processing, quality control, normalization and visualization of single-cell RNA-seq data in R. *Bioinformatics* (2017).
555 doi:10.1093/bioinformatics/btw777
9. Scialdone, A. *et al.* Resolving early mesoderm diversification through single-cell expression profiling. *Nature* (2016). doi:10.1038/nature18633
10. Kharchenko, P. V., Silberstein, L. & Scadden, D. T. Bayesian approach to single-cell differential expression analysis. *Nat. Methods* (2014). doi:10.1038/nmeth.2967

560

Reviewers' comments:

Reviewer #1 (Remarks to the Author):

Revisions helped but the manuscript is still quite narrow and small scale, interpretations difficult and ambivalent, and the overall impact is limited.

The new section on epigenetic aging is quite interesting but again, there is a series of issues:

1. "Epigenetic clock ticks more slowly in adult stem cells". Really? The actual animal ages were 1.5 months (=6 weeks) and 26 months; stem cell age prediction showed 9 weeks and 10-11 weeks, respectively. So, it was accelerated aging in the young adult animals and decelerated in the old ones.
2. Then a new challenge became an explanation which makes compatible slow epigenetic aging with all other age-related changes in the old stem cells. DNA methylation in stem cells is "aging" or not?
3. The interpretation provided in Discussion is confusing. My understanding is that the authors believe that the predictor primarily measured cell count changes in the aging tissue (external clock, according to Horvath), which did not work on purified homogenous stem cells (internal clock). If yes, this needs to be explicitly stated. Remains unclear if age discrepancy is it due to limitations of the algorithm or it is genuine deceleration (but see #1)?

Reviewer #2 (Remarks to the Author):

The authors have now addressed all my concerns, and as far as I can tell those of the other reviewers, in the revised version of the manuscript.

Reviewer #3 (Remarks to the Author):

The authors have addressed my previous critiques, and I have no further concerns with the manuscript.

Reviewer #4 (Remarks to the Author):

In this revised manuscript, Irene Hernando-Herraez et al. have addressed some of my concerns, for instance, my previous main point #2 regarding sc-RNaseq data quality control. However, most of my other comments are still left without reasonable rebuttal. The authors also failed to provide any further experimental evidence to support a few important issues that I had pointed out in my previous review. Please check below for details:

For previous main point #1, I agree that the genetic knockouts and epigenetic editing are particularly challenging. However, it is at least feasible and time-permitting to detect the expressions of several important genes by using molecular biological technique, such as FISH and immunofluorescent staining. Through this, the data presented in this manuscript will be much more conclusive. It is not advisable just to tone-down the already weak points.

For previous main point #3, I don't think cell proliferation can really be a confounding factor, as there are several ways to regress out the effect of cell cycle in data analysis if the authors want. Neither I think differentiation states can be confounding factor but actually very interesting point (see exactly the same comment from reviewer 1). Furthermore, I believe time and costs are not

good reasons to refuse to perform a key experiment. Actually, even only scRNAseq (without scBSseq) of total PAX7-positive cells should be able to acquire much more information than the current manuscript demonstrates.

For previous main point #4, it is hard for me to buy it that the only category that is significant among the genes with increased transcriptional variability with age is "signaling receptor activity". If this is true as the authors claim, I would suspect that those genes presented in the paper are subjectively selected, hence double about the reliability of the conclusion based on the data Fig. 1C and 2A.

Another comment regarding authors' rebuttal that needs to ponder is that the authors may believe too deeply in the meaning of p value, for instance in main point 3 and minor point 2. Using p value should be very careful and rather than making scientific conclusions based solely on p value. All statistical experiments and the choice made in the calculations should be reasonably interpreted, otherwise the results are likely to look unreliable.

Reviewer #1 (Remarks to the Author):

Revisions helped but the manuscript is still quite narrow and small scale, interpretations difficult and ambivalent, and the overall impact is limited.

The new section on epigenetic aging is quite interesting but again, there is a series of issues:

1. "Epigenetic clock ticks more slowly in adult stem cells". Really? The actual animal ages were 1.5 months (=6 weeks) and 26 months; stem cell age prediction showed 9 weeks and 10-11 weeks, respectively. So, it was accelerated aging in the young adult animals and decelerated in the old ones.

We thank the Reviewer for the comment and agree that the phrasing was unclear. The predicted ages for muscle stem cells for all our samples are very close to each other and the observed differences fall within the error range for the model (predicted ages 9.2 and 9.3 weeks in the young samples and 10.2 and 11.6 weeks in the old ones, error in the training set 3.2 weeks). However, in order to investigate this point further, we have now estimated the impact of subsampling the merged single cells, to get an idea of the ages of the single cells and the error of the modelling. We performed 100 permutations of the single cells by randomly removing 5% of the cells in each permutation and calculated the epigenetic age of each subset. Given the coverage of the single cells we used only the permutations with at least 4x coverage on each of the sites used in the clock. These results show a significant increase in the epigenetic age of old samples; however, the predicted ages are still very different from the chronological age of the samples (Figure 3A). These results suggest two things: firstly, cell type and probably differentiation potential are important factors contributing to the model. In this sense, adult stem cells would have a different epigenetic age than postmitotic cells. Based on this hypothesis, the epigenetic clock would capture, at least in part, the differences in cell type composition in the ageing tissue (external clock). Secondly, the slight increase of the epigenetic age observed in old samples suggests intrinsic changes at the cellular level, in the stem cell population (internal clock). Therefore, the epigenetic clock would then be a combination of these internal and external factors. Technical factors, such as the datasets used to train the model, may also contribute to these predictions. We have now revised the manuscript to achieve greater clarity and avoid confusion (Lines 151-159; 227-242) and added the new analyses in Figure 3A.

2. Then a new challenge became an explanation which makes compatible slow epigenetic aging with all other age-related changes in the old stem cells. DNA methylation in stem cells is "aging" or not?

We thank the Reviewer for the comment. In our study, we have observed changes at the epigenetic level with age, some of them associated with transcriptional changes, suggesting that DNA methylation in stem cells is "ageing". However, this is not well captured in the epigenetic clock (see previous comment). We have now explicitly discussed these points in the manuscript (Lines 151-159; 227-242)

and we conclude that stem cells do show significant DNA methylation alterations with ageing but are not strictly “ageing” as defined by the epigenetic clock.

3. The interpretation provided in Discussion is confusing. My understanding is that the authors believe that the predictor primarily measured cell count changes in the aging tissue (external clock, according to Horvath), which did not work on purified homogenous stem cells (internal clock). If yes, this needs to be explicitly stated. Remains unclear if age discrepancy is it due to limitations of the algorithm or it is genuine deceleration (but see #1)?

We thank the Reviewer for the comment and agree that the phrasing was unclear. Following the Reviewer’s comment, we have now clearly discussed this interpretation in the manuscript (see previous comments).

Reviewer #2 (Remarks to the Author):

The authors have now addressed all my concerns, and as far as I can tell those of the other reviewers, in the revised version of the manuscript.

Reviewer #3 (Remarks to the Author):

The authors have addressed my previous critiques, and I have no further concerns with the manuscript.

Reviewer #4 (Remarks to the Author):

In this revised manuscript, Irene Hernando-Herraez et al. have addressed some of my concerns, for instance, my previous main point #2 regarding sc-RNAseq data quality control. However, most of my others comments are still left without reasonable rebuttal. The authors also failed to provide any further experimental evidence to support a few important issues that I had pointed out in my previous review. Please check below for details:

For previous main point #1, I agree that the genetic knockouts and epigenetic editing are particularly challenging. However, it is at least feasible and time-permitting to detect the expressions of several important genes by using molecular biological technique, such as FISH and immunofluorescent staining. Through this, the data presented in this manuscript will be much more conclusive. it is not advisable just to toned-down the already weak points.

We thank the Reviewer for the comment. Following the Reviewer's suggestion, we have validated the expression pattern of genes showing expression frequency differences with age using a different methodology and at the protein level. We selected Itgb1 and Cdh15 as antibodies are available against these proteins and validated our transcriptional findings by immunocytochemistry on muscle stem cells. As a control we quantified immunofluorescence intensity of nGFP (*Tg:Pax7-nGFP*, proxy for Pax7 expression) and observed no differences in nGFP between muscle stem cells from young and old mice. However, while all analysed young muscle stem cells expressed Cdh15 or Itgb1, a few old muscle stem cells were showing low to no expression of Cdh15 (20%, 8 out of 40 analysed cells) or Itgb1 (12.5%, 5 out of 40 analysed cells). These observations confirm at the protein level the increased expression variability with age observed by scM&T-seq. We have now included these results in the manuscript (Lines 116-125, Figure S8).

For previous main point #3, I don't think cell proliferation can really be a confounding factor, as there are several ways to regress out the effect of cell cycle in data analysis if the authors want. Neither I think differentiation states can be confounding factor but actually very interesting point (see exactly the same comment from reviewer 1). Furthermore, I believe time and costs are not good reasons to refuse to perform a key experiment. Actually, even only scRNAseq (without scBSseq) of total PAX7-positive cells should be able to acquire much more information than the current manuscript demonstrates.

We thank the Reviewer for the comment. We have now sequenced by scRNA-seq the total Pax7 positive population from two young and two old individuals, 96 cells per sample. A total of 147 cells passed quality control using the same parameters used for our Pax7^{Hi} population (Figure S16, Figure S18). Despite the relatively low numbers, we observed consistent results between the Pax7^{Hi} and Pax7-total populations. In particular, we observed similar cell cycle phase (Figure #1), absence of clear clustering or subclusters (Figure #2), increase of transcriptional variability with age (Figure S7 top) and similar gene expression patterns of our example genes (Figure S7 bottom). We have now included these results in the manuscript (Lines 114-116, Figure S7).

Figure #1. Cell cycle stage of Pax7-total cells.

Scatter plots of predicted phases of the cell cycle (G1 score and G2M score) for single cells (grey circles) from young and old mice.

Figure #2: The two-dimensional outputs of t-SNE using highly variable genes from different scRNA-seq datasets (top: Pax7^{Hi} and Pax7-total, bottom: Pax7-total). Different plots show different values of perplexity. Number of iterations=10000.

For previous main point #4, it is hard for me to buy it that the only category that is significant among the genes with increased transcriptional variability with age is "signaling receptor activity". If this is true as the authors claim, I would suspect that those genes presented in the paper are subjectively selected, hence double about the reliability of the conclusion based on the data Fig. 1C and 2A.

We thank the Reviewer for the comment; we agree that the explanation could be misleading. The genes presented in Figure 1C were selected based on current literature¹. For example, *Myod* and *Myf5* were included because they are well known myogenic markers and *Spry* is a key quiescence factor^{1,2}.

We have now performed Gene Ontology analysis on upregulated and downregulated genes with age (Figure S3)³. Among the terms associated with the upregulated genes we noted that some of them, such as translation or peptide biosynthetic process, were also associated with the "older" phenotype of aged cells (Figure 2B).

In addition, we have also now performed Gene Ontology analysis on the genes with decreased expression frequency and observed extracellular matrix structure and organization are one of the top categories (Figure S5). In addition, we noted that these categories are also observed in the "younger" phenotype of aged cells (Figure 2A). Based on these observations, we included examples (Figure 2A) of genes showing decreased expression frequency (Table S3) and associated with these GO categories such as *Itgb1* (expression frequencies: young 0.79, old 0.59), *Dag1* (expression frequencies: young 0.96, old 0.65), or *Cdh15* (expression frequencies: young 0.62, old 0.41). *Itgb1* was also included in Figure 2A because its activation has been shown to be impaired in aged muscle stem cells⁴.

When restricting the analysis to genes that have increased variability with age by correcting by mean expression levels, the only category that is significant after applying FDR correction is 'signaling receptor activity'. We would like to mention that this analysis is much more restrictive and requires both genes to be expressed in young and old samples (average normalized read count of at least 10). Importantly, *Dag1* is also one of the top genes (top 50) with increased variability with age using this method (Table S3). Following the Reviewer's comment, we have now included these analyses in the manuscript (Lines 82; 106. Fig. S3 and Fig. S5).

Another comment regarding authors' rebuttal that needs to ponder is that the authors may believe too deeply in the meaning of p value, for instance in main point 3 and minor point 2. Using p value should be very careful and rather than making scientific conclusions based solely on p value. All statistical experiments and the choice made in the calculations should be reasonably interpreted, otherwise the results are likely to look unreliable.

Although the p-value is widely used by the scientific community, we agree that it may not be proof of strong evidence. When analysing cell cycle entry through in vivo BrdU uptake (previous main point 3), we observed small to no differences (Fisher exact test) between ages whether analysing the total Pax7

population or Pax7^{hi} subpopulation. However, we agree that the biological relevance of these measures cannot be inferred from the calculated p-values.

Regarding minor point 2 about the differential expression analysis (Figure 1C), we have now clarified in the manuscript that in some cases the differences may be small (line 82). However, we would like to mention that this analysis is based on one of the most common methods for measuring differential expression using single cell data⁵.

References

1. Brack, A. S. & Rando, T. A. Tissue-specific stem cells: lessons from the skeletal muscle satellite cell. *Cell Stem Cell* **10**, 504–514 (2012).
2. Bigot, A. *et al.* Age-associated methylation suppresses SPRY1, leading to a failure of re- quiescence and loss of the reserve stem cell pool in elderly muscle. 1172–1182 (2015). doi:10.1016/j.celrep.2015.09.067
3. Eden, E., Navon, R., Steinfeld, I., Lipson, D. & Yakhini, Z. GOrilla: A tool for discovery and visualization of enriched GO terms in ranked gene lists. *BMC Bioinformatics* (2009). doi:10.1186/1471-2105-10-48
4. Brennecke, P. *et al.* Accounting for technical noise in single-cell RNA-seq experiments. *Nat. Methods* **10**, 1093–1095 (2013).
5. Kharchenko, P. V., Silberstein, L. & Scadden, D. T. Bayesian approach to single-cell differential expression analysis. *Nat. Methods* (2014). doi:10.1038/nmeth.2967

REVIEWERS' COMMENTS:

Reviewer #4 (Remarks to the Author):

In the revised version of the manuscript, The authors have addressed my previous major concerns. Therefore we consider this works suitable for publication in Nature Communications.